# Normative evidence weighing and accumulation in correlated environments

**Nathan Tardiff[1,2,3]\*, Jiwon Kang[3], Joshua I Gold[3]**

[1]Department of Otorhinolaryngology, Perelman School of Medicine, University of Pennsylvania, Philadelphia, United States; [2]Department of Psychology, New York University, New York, United States; [3]Department of Neuroscience, Perelman School of Medicine, University of Pennsylvania, Philadelphia, United States

## eLife Assessment

This **important** work combines theory and experiment to demonstrate **convincingly** how humans make decisions about sequences of pairs of correlated observations. The proposed model for evidence integration in correlated environments will be of use for the study of decision-making.

**Abstract** The brain forms certain deliberative decisions following normative principles related to how sensory observations are weighed and accumulated over time. Previously we showed that these principles can account for how people adapt their decisions to the temporal dynamics of the observations (Glaze et al., 2015). Here, we show that this adaptability extends to accounting for correlations in the observations, which can have a dramatic impact on the weight of evidence provided by those observations. We tested online human participants on a novel visual-discrimination task with pairwise-correlated observations. With minimal training, the participants adapted to uncued, trial-by-trial changes in the correlations and produced decisions based on an approximately normative weighing and accumulation of evidence. The results highlight the robustness of our brain's ability to process sensory observations with respect to not just their physical features but also the weight of evidence they provide for a given decision.

**\*For correspondence:**
ntardiff@sas.upenn.edu

## Introduction

In their efforts to break the Enigma code during World War II, Alan Turing and his colleagues at Bletchley Park recognized the importance of the concept of a 'weight of evidence' for making decisions: noisy or ambiguous evidence is most useful when the influence or weight it has on the ultimate decision depends on its uncertainty. For the case of two alternatives, they used a weight of evidence in the form of the logarithm of the likelihood ratio (i.e., the ratio of the likelihoods of each of the two alternative hypotheses, given the observations), or logLR. The logLR later became central to the sequential probability ratio test (SPRT), which was proven to provide certain optimal balances between the speed and accuracy of such decisions (*Barnard, 1946*; *Wald, 1947*; *Wald and Wolfowitz, 1948*). Recognizing the general nature of this formulation, Turing and colleagues noted that the logLR would be "an important aid to human reasoning and … eventually improve the judgment of doctors, lawyers, and other citizens" (*Good, 1979*).

The logLR has since become ubiquitous in models of decision-making. Examples include sequential-sampling models related to the SPRT like the drift-diffusion model (DDM), which can capture many behavioral and neural features of human and animal decision-making for a broad range of tasks (*Gold and Shadlen, 2007*; *Smith and Ratcliff, 2004*). These models typically assume that the decision is formed by accumulating over time evidence from statistically independent observations until reaching

a threshold value, or bound. The magnitude of this bound governs a trade-off between decision speed and accuracy (lower bounds emphasize speed, higher bounds emphasize accuracy; *Heitz, 2014*). The weight of evidence is computed as a scaled version of each observation (the scaling can be applied to the observations or to the bound, which are mathematically equivalent; *Green and Swets, 1966*) to form the logLR.

However, it is often unclear if and when decision-makers use the logLR or instead rely on approximations or other heuristics (*Brown et al., 2009*; *Hanks et al., 2011*; *Ratcliff et al., 2016*; *Ratcliff and McKoon, 2008*). A major complication is that the logLR can be difficult to compute because it depends on detailed knowledge of the statistical properties of the observations. One such property is the signal-to-noise ratio (SNR) of the observations. When the SNR is not directly accessible to the decision-maker (e.g., when signal strength is varied randomly from trial to trial, as is common for many laboratory tasks), it can be approximated using surrogates like elapsed decision time to help calibrate the weight of evidence (*Drugowitsch et al., 2012*; *Hanks et al., 2011*).

Another statistical property that has received less attention is non-independence of the observations, which can have substantial effects on how those observations should be weighed to form effective decisions (*Figure 1*). If not accounted for appropriately, these effects can lead to over- or underestimates of the weight of available evidence and suboptimal decisions. Such suboptimalities have real-world consequences. For example, misestimation of correlation patterns in mortgage defaults is thought to have played a role in triggering the global financial crisis of 2008 (*Salmon, 2009*). Neglecting correlations can contribute to false beliefs and ideological extremeness in social and political settings (*Denter et al., 2021*; *Glaeser and Sunstein, 2009*; *Levy et al., 2022*; *Ortoleva and Snowberg, 2015*). Likewise, correlations in the physical environment should, in principle, be leveraged to support perception (*Geisler, 2008*; *Parise, 2016*). Yet whether and how people account for correlations when making perceptual decisions is not well understood.

The goal of this preregistered study (https://osf.io/qj92c) was to test how humans form simple perceptual decisions based on observations with different degrees of correlation. We and others previously showed that both theoretically optimal (i.e., an ideal observer that maximizes decision accuracy) and human observers flexibly adjust how evidence is accumulated over time to account for the temporal dynamics of the sequentially presented observations (*Glaze et al., 2015*; *Veliz-Cuba et al., 2016*). Here, we assess how both ideal and human observers weigh and accumulate evidence that is based on pairs of correlated observations (*Figure 2*; we do not consider other forms of correlation, such as over time). Under these conditions, normative decisions use an accumulate-to-bound process that has been scaled appropriately to produce a correlation-dependent weight of evidence. As we detail below, we found that people tend to follow these normative principles, accounting appropriately for the correlations (albeit based on slight misestimates of correlation magnitude) and demonstrating the robustness and flexibility with which our brains can appropriately weigh and accumulate evidence when making simple decisions.

## Results

We tested 100 online participants performing a novel task that required them to form simple decisions about which of two latent sources generated the observed visual stimuli in the presence of different correlation structures in the stimuli (*Figure 2*). The task design was based on principles illustrated in *Figure 1*: the normative weight of evidence for the identity of a source of paired observations from correlated, Gaussian random variables depends systematically on the sign and magnitude of the correlation (*Figure 1b*). For this case, negative pairwise correlations provide, on average, stronger evidence with increasing correlation magnitude because less overlap of the generative source distributions allows them to be more cleanly separated by the decision boundary (*Figure 1c*, left inset). Positive pairwise correlations provide, on average, weaker evidence with increasing correlation magnitude, because more overlap of the generative source distributions causes them to be less cleanly separated by the decision boundary (*Figure 1c*, right inset).

Participants reported the generative source (left or right) of noisy observations, depicted onscreen as the position of stars along a horizontal line (*Figure 2a*). Observations were presented in pairs. Each element of the pair had the same mean value across samples from the generative source, while the noise correlation within pairs was manipulated on a per-trial basis. We assigned each participant to a correlation-magnitude group ($|\rho|$=0.2, 0.4, 0.6, 0.8; 25 participants per group) in which the

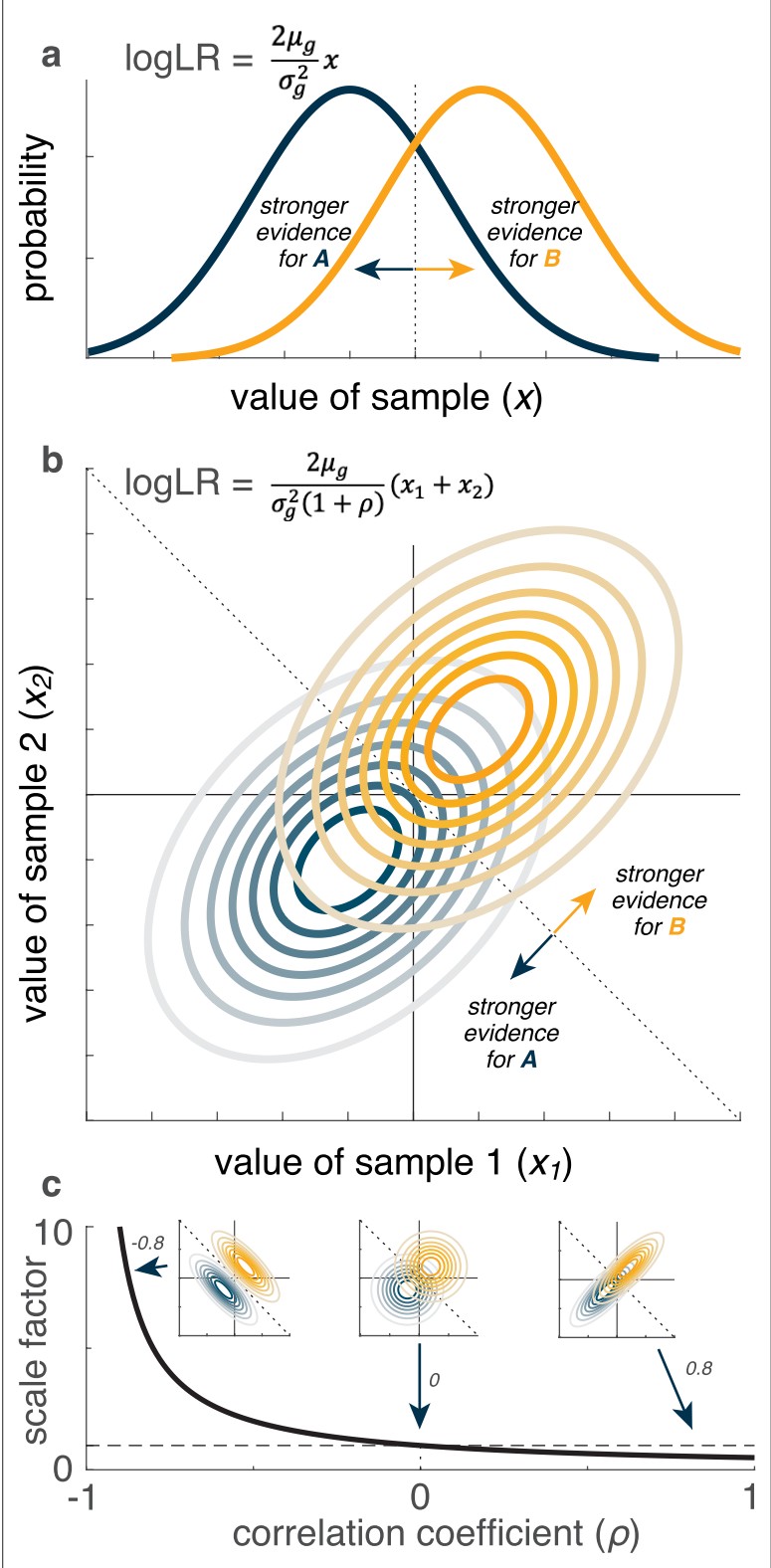

**Figure 1.** Illustration of how pairwise correlations can affect the weight of evidence (logLR) for the generative source of an observation. (**a**) Computing the logLR when the observation ($x$) is a single sample from one of two one-dimensional Gaussian distributions (labeled $A$ and $B$), with means $\pm\mu_g$ and equal variances ($\sigma_g^2$) (**Gold and Shadlen, 2001**). (**b**) Computing the logLR when the observation ($x_1, x_2$) is a pair of samples from one of two pairs of one-dimensional Gaussian distributions (labeled $A$ and $B$), with means $\pm\mu_g$, equal variances ($\sigma_g^2$), and

*Figure 1 continued on next page*

*Figure 1 continued*

correlation between the two Gaussians = $\rho$. (**c**) The normative, correlation-dependent scaling of the weight of evidence ($\frac{1}{1+\rho}$ term in **b**) of the observation plotted as a function of the correlation. The dashed horizontal line corresponds to scale factor = 1, which occurs at $\rho$ = 0. The insets show three example pairs of distributions with different correlations, as indicated. The dotted lines in (**a**, **b**), and the insets in (**c**) indicate the optimal decision boundary separating evidence for *A* versus *B*.

pairwise correlation on a given trial was drawn from three conditions: $\rho_-$, 0, or $\rho_+$. We equated task difficulty across participants by calibrating the means of the generative distributions (see 'Materials and methods'). We interleaved randomly the three correlation conditions with the two sources (left, right) and two levels of task difficulty (low, high), for 12 total conditions for each correlation-magnitude group.

Crucially, we adjusted the means of the generative distributions to ensure that the expected logLR (which we term the objective evidence strength) was constant across correlation conditions (*Figure 2b and c*). For example, because negative correlations increase logLR for the same generative mean and standard deviation (*Figure 1c*), we used smaller differences in means for the negative-correlation conditions than for the zero-correlation conditions. As a result, we expected participants who made decisions by weighing the evidence according to the true logLR to produce identical distributions of choices and response times (RTs) across correlation conditions. In contrast, we expected participants who ignored the correlations to underweigh the evidence provided by negative-correlation pairs and overweigh the evidence provided by positive-correlation pairs, which would affect RTs and/or choices. We further expected strategies between these two extremes to have more mixed effects on behavior, as we detail below.

## Human response times are influenced by correlated observations

The example participant in *Figure 3a* exhibited behavioral patterns that were illustrative of the overall trends we observed. Specifically, their choice accuracy was affected by objective evidence strength (higher accuracy for stronger evidence) but not correlation (this participant was tested using correlation values of −0.6, 0.0, and 0.6). In contrast, their RTs were affected by both evidence strength and correlation, including faster responses on correct trials using stronger evidence and more positive correlations.

Likewise, across our sample of participants choices depended strongly on evidence strength but not on correlation (*Figure 3b*). Logistic models fit to individual participant's evidence-strength-dependent psychometric data demonstrated no benefit to fitting separate models per correlation condition versus a single model fit jointly to all three correlation conditions (mean ΔAIC=−4.14, protected exceedance probability [PEP]=1.0 in favor of the joint model). This result also held true at each correlation magnitude individually (all mean ΔAIC<−2.0, all PEP>0.8).

In contrast, RTs were affected by both evidence strength and correlation, with a tendency of participants to respond faster for stronger evidence and more-positive correlations (*Figure 3b*). A linear mixed-effects model fit to median RTs from correct trials confirmed these observations, indicating effects on RT of evidence strength ($F$(1,98.00)=174.24, p<0.001), the sign of the correlation (negative, zero, positive) within participants ($F$(2,131.56)=219.96, p<0.001), and the interaction between the sign of the correlation and its magnitude between participants ($F$(2,131.56)=81.04, p<0.001). That is, the effects of correlations on correct RTs were more pronounced in participants tested using stronger correlations. Similar effects were also present on error trials (evidence strength: $F$(1,960.54)=19.21, p<0.001), sign of correlation: ($F$(2,234.74)=58.41, p<0.001), and correlation sign × magnitude ($F$(2,233.48)=13.50, p<0.001).

In short, the patterns of choice data that we observed did not vary across correlation conditions, as expected for decisions that took into account the correlations in the observations. However, the RT data imply that the influence of the correlations on decisions was not optimal (i.e., was not equivalent to using a weight of evidence based on the true logLR) because if it were, the RTs also would not vary by correlation condition. These findings leave open a broad range of possible weighing strategies between the two extremes of an ideal observer and a naïve observer who ignores the correlations (i.e., assumes independence). The analyses detailed below aimed to more precisely identify where in that range our participants' strategies fell.

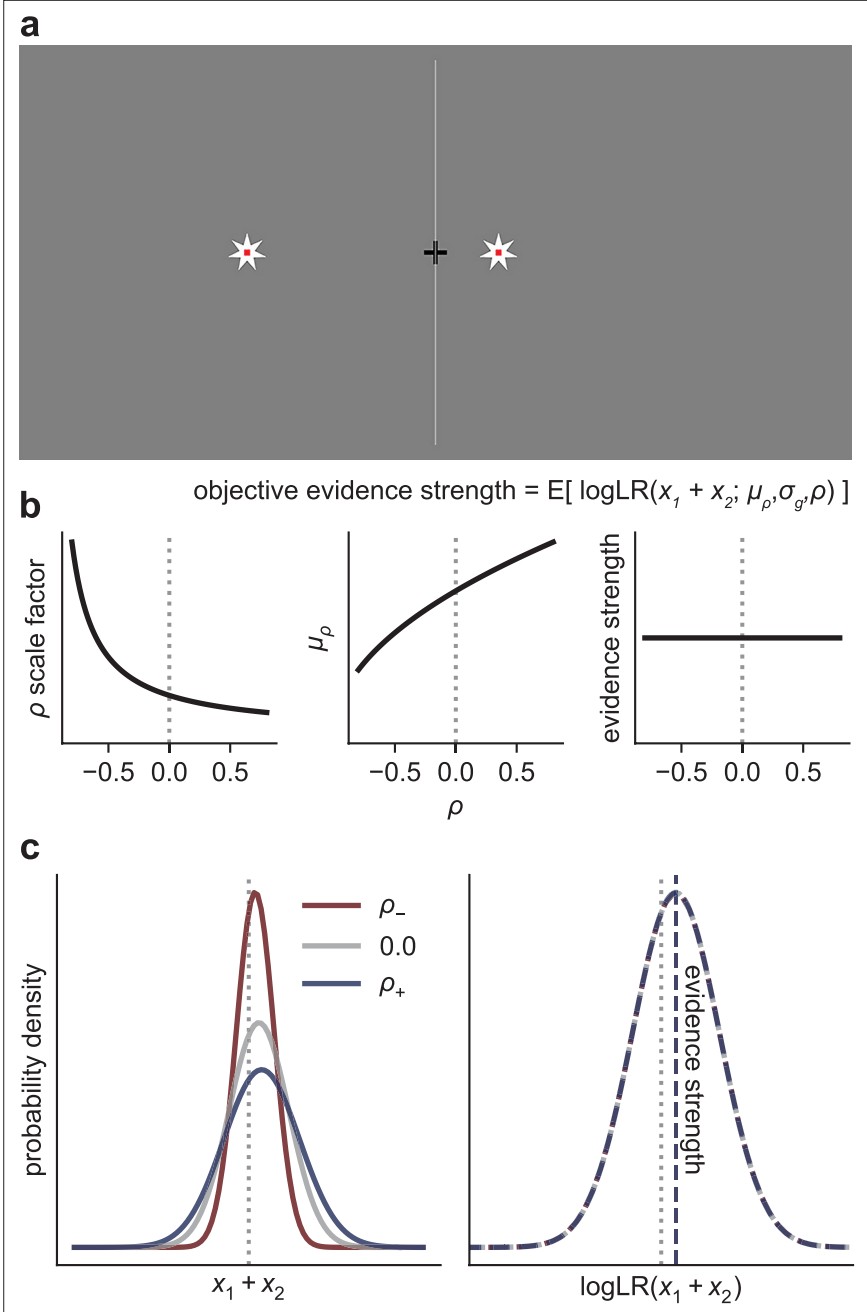

**Figure 2.** Task. (**a**) Human observers viewed pairs of stars (updated every 0.2 s) and were asked to decide whether the stars were generated by a source on the left or right side of the screen. An example star pair is shown. The horizontal position of each star pair was drawn from a bivariate Gaussian distribution, with a mean and correlation that varied from trial-to-trial. (**b**) Because the normative correlation-dependent scale factor that converts observations to evidence (logLR) increases as the correlation decreases, we manipulated the mean of the generative distribution such that the expected logLR (objective evidence strength) was fixed across correlation conditions. (**c**) The generative distributions of the sum of individual star pairs, for three example correlation conditions. Decreasing the correlation has the effect of decreasing the standard deviation of the sum distribution. By adjusting each correlation-specific generative mean ($\mu_\rho$) in proportion to the correlation-dependent change in the standard deviation from the zero-correlation condition (i.e., $\mu_\rho = \mu_0\sqrt{1+\rho}$), the true logLR distribution (i.e., of an ideal observer) is invariant to the correlation, and thus evidence strength remains fixed. Note that the sum-of-pairs distribution is equivalent to the bivariate distribution for the purposes of computing the logLR (see 'Materials and methods').

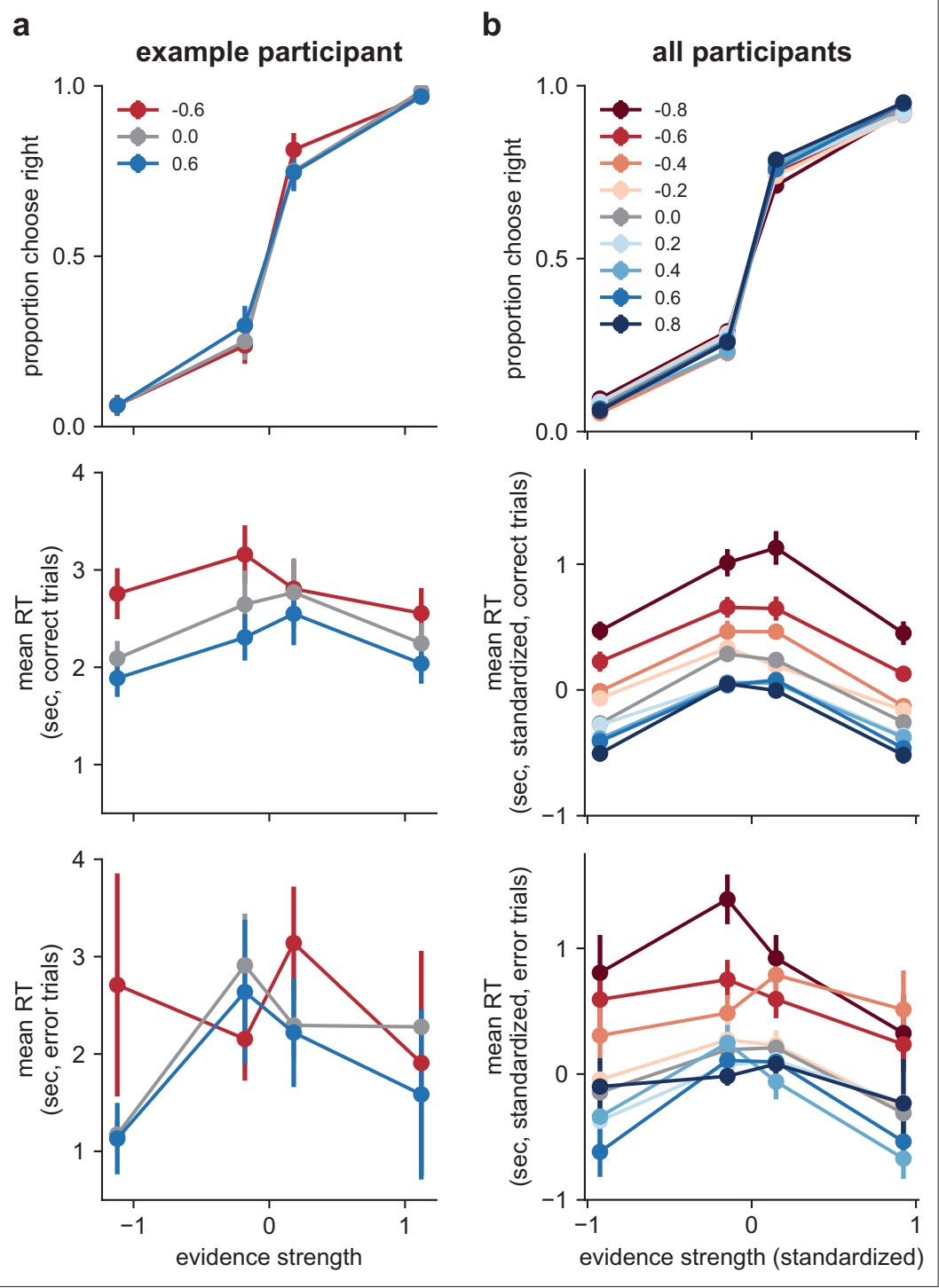

**Figure 3.** Effects of correlations on choice and response time (RT). (**a**) Data from an example participant from the 0.6 correlation-magnitude group. Top: choices plotted as a function of evidence strength (abscissa) and correlation condition (see legend). Middle, bottom: mean RTs for correct and error trials, respectively. Error bars are within-participant standard errors of the mean (SEM). (**b**) Same as (**a**), but data are averaged across all participants (25 per correlation-magnitude group). Evidence strength was standardized to equal the mean evidence strength (expected logLR) for each condition, across participants. RT was standardized by subtracting each participant's mean RT in the zero-correlation condition, separately for correct and error trials. Points and error bars are across-participant means and SEMs, respectively.

## RTs are consistent with a decision bound on approximate logLR

We analyzed the RT data in more detail, assuming that, like for the DDM and SPRT, the decision was formed by accumulating evidence over time until reaching one of two fixed bounds. This process governs both the choice (which bound is reached first) and RT (when the bound is reached). We considered evidence with weights determined by three different scale factors (i.e., the value by which to multiply each star position or, equivalently, divide each decision bound to govern the weight of evidence for a given set of task conditions): (1) 'unscaled' evidence was taken directly as each star position; (2) 'naïve' evidence was scaled by the generative mean, $\mu_g$, of a pair of samples, which produces a weight of evidence equivalent to a mis-specified logLR that ignores the correlations; and (3) 'true' evidence was scaled by $\frac{\mu_g}{1+\rho}$, which takes into account the correlations and produces a weight of evidence equivalent to the true logLR.

Because we designed the task to present stimuli with equal expected logLR (objective evidence strength) across correlation conditions, decisions based on an accumulation of the true logLR to a fixed bound would have similar mean RTs across correlation conditions. In contrast, decisions based on an accumulation of the unscaled or naïve logLR would have different effects for positive versus negative correlations. Ignoring positive correlations is equivalent to ignoring redundancies in the observations, which would lead to overweighing the evidence and thus reaching the bound more quickly, corresponding to shorter RTs. Ignoring negative correlations is equivalent to ignoring synergies in the observations, which would lead to underweighing the evidence and thus reaching the bound less quickly, corresponding to longer RTs (*Figure 1c*).

The participants had RTs that were, on average, either relatively constant or slightly decreasing as a function of increasing correlations, particularly for larger correlations (*Figure 4a and b*). These trends were not consistent with a decision process that used a fixed bound that ignored correlations (either with or without additional scaling). They also were not completely consistent with a decision process that used a fixed bound on the true logLR because of the dependency of RT on the correlations. Instead, these results could be matched qualitatively to simulations that made decisions based on an approximation of logLR computed using underestimates of the correlation-dependent scale factor (*Figure 4c*). We examined this idea more quantitatively using model fitting, detailed in the next section.

## Correlation-dependent adjustments in a drift-diffusion model

To better understand how the participants formed correlation-dependent decisions, we developed variants of the DDM that can account for pairwise-correlated observations. The DDM jointly accounts for choices and RTs according to a process that accumulates noisy evidence over time until reaching a decision bound (*Figure 5a*). The model includes two primary components that govern the decision process. The *drift rate* governs the average rate of information accumulation (*Palmer et al., 2005*; *Ratcliff and McKoon, 2008*). This term typically depends on the product of the strength or quality of the sensory observations (generally varied via the mean, or signal, of the observation distribution, $\mu_g$) and the decision-maker's sensitivity to those observations (the fit parameter $k$; i.e., *drift rate*$(\rho) = k\mu$). The *bound height* governs the decision criterion, or rule, which corresponds to the amount of evidence required to make a decision and controls the trade-off between decision speed and accuracy (*Heitz, 2014*). We used a single fit parameter representing symmetric bounds (*bound height = B*). Following previous approaches, we also included an additional parameter ($t_B$) to govern the rate of a linear 'collapse' of the bounds over time, which can serve to calibrate the weight of evidence when the objective evidence strength is varied from trial to trial (*Drugowitsch et al., 2012*; *Hanks et al., 2011*).

For our task, changes in $\rho$ affected the standard deviation of the distribution of sum-of-pairs observations, $\sigma_\rho$ (i.e., $\sigma_\rho \propto \sqrt{1+\rho}$). The DDM typically assumes that this observation variability is captured by a noise term (the 'diffusion' part of 'drift-diffusion') that is used to normalize both the drift rate and bound height (*Palmer et al., 2005*). Conventionally, this scaling is implicit: by assuming that the diffusion is constant across conditions, this scaling is simply subsumed into the drift-rate and bound-height parameters, and the diffusion parameter is set to one. In contrast, our models accounted for the correlation-dependence of this diffusion term explicitly so that we could fit a single model to the correlation-dependent data from each participant. Critically, this correlation-dependent diffusion was a component of the internal decision process and not the external task features. Thus, we formulated it using a pair of free parameters that represented subjective estimates of the negative ($\hat{\rho}_{SD-}$)

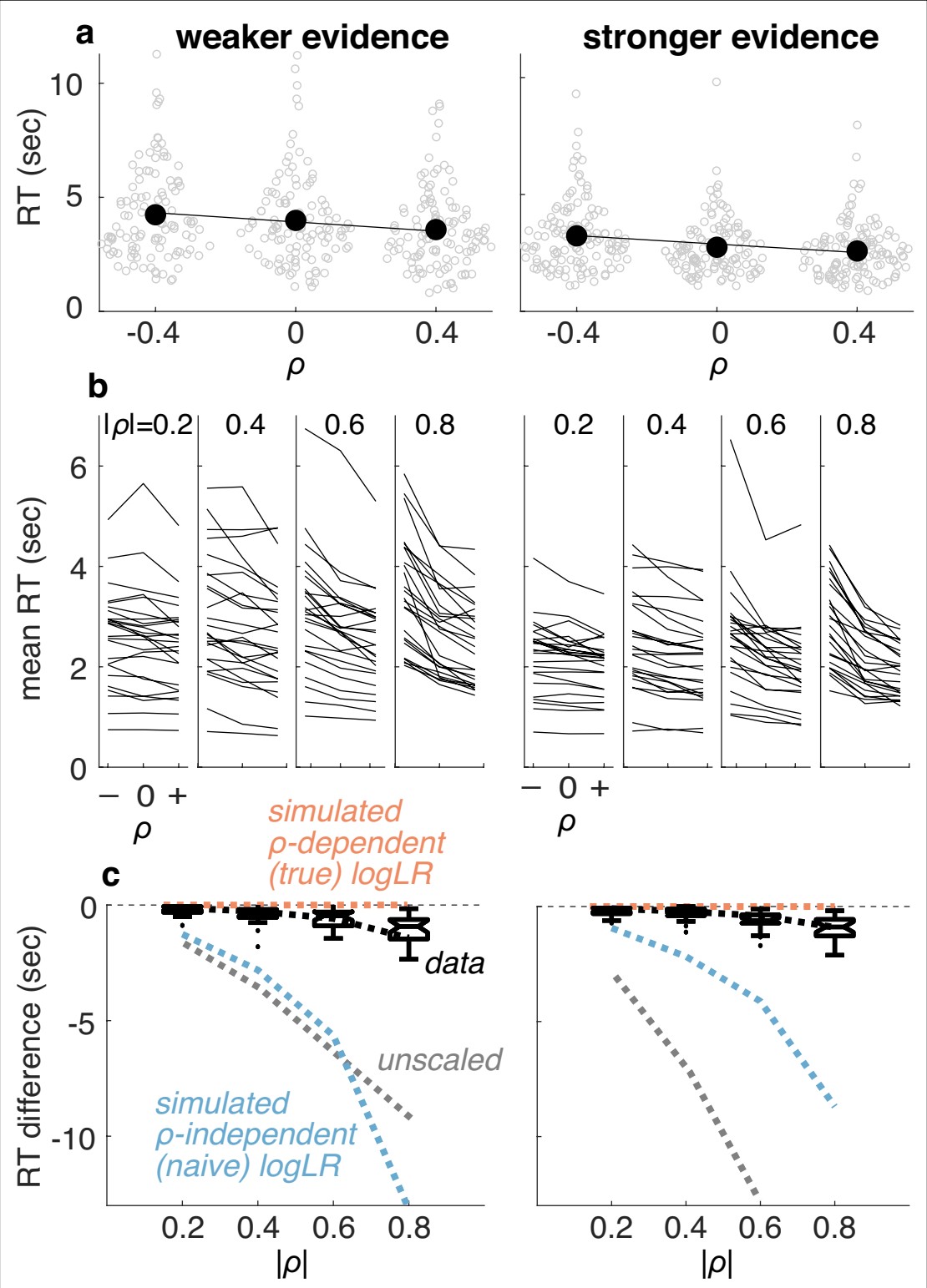

**Figure 4.** Response times (RTs) were consistent with a bound on (approximate) logLR. (**a**) RTs measured from an example participant for the weaker (left) and stronger (right) evidence conditions. Unfilled points are data from individual trials. Filled points are means, lines are linear fits to those means. (**b**) Summary of mean RT versus correlation for all participants and conditions. Correlation-magnitude group is indicated at the top of each panel. Lines are data from individual participants. (**c**) Summary of differences in mean RT between the positive- versus negative-correlation condition for individual participants (as in **a**). Box-and-whisker plots show median, interquartile range, 90th percentiles, and outliers as a function of correlation-magnitude group. Colored lines are predicted relationships for decisions based on an accumulation of evidence to a fixed bound, where the weight

*Figure 4 continued on next page*

*Figure 4 continued*

of evidence was computed as unscaled, correlation-independent (naïve), or correlation-dependent (true) logLR. The data are roughly consistent with decision processes that, on average, used a correlation-dependent logLR but based on a slight underestimate of the correlation-dependent scale factor (computed using $\frac{\mu_g}{1+0.9\rho}$; black dashed lines).

The online version of this article includes the following source data for figure 4:

**Source code 1.** Code for generating *Figure 4*.

**Source code 2.** Code for generating *Figure 4c*.

**Source data 1.** Behavioral data used to create the figures generated by *Figure 4—source code 1*, *Figure 7—figure supplement 2—source code 1*, and *Figure 8—source code 1*.

and positive ($\hat{\rho}_{SD+}$) correlations affecting the standard deviation of the internal noise distribution, as follows.

For the drift rate, dividing by the component of the noise distribution ($\sigma_\rho$) that depended on the appropriate $\hat{\rho}_{SD}$ yields:

$$drift\,rate(\rho) = k\frac{\mu_\rho}{\sigma_\rho} = \frac{k_0}{\sqrt{1+\hat{\rho}_{SD}}}\mu_0\sqrt{1+\rho},$$

where $k_0$ is the drift-rate parameter for the zero-correlation condition. Here, we also express $\mu_\rho$ (the 'drift' part of 'drift-diffusion') as the $\rho$-dependent generative mean that for our task was set to $\mu_0\sqrt{1+\rho}$ (see *Figure 2*). This formulation allows for both linear and nonlinear relationships between objective stimulus properties (governed by task parameters $\mu_0$ and $\rho$) and subjective stimulus strength (governed by fit parameters $k_0$ and $\hat{\rho}_{SD}$). As such, it leaves open the possibility of a variety of participant-specific suboptimalities in the decision process, which are often found for decisions requiring the simultaneous accumulation of evidence from multiple sources (*Kang et al., 2021*; *Luyckx et al., 2020*; *Usher et al., 2019*; *Wyart et al., 2015*; cf. *Rangelov et al., 2024*).

For the bound height, dividing by the component of the noise distribution ($\sigma_\rho$) that depended on the subjective correlation yielded a similar dependence on $\sqrt{1+\hat{\rho}_{SD}}$. However, the correlation dependence of the bound was further complicated by an additional scaling to convert the sum-of-pairs observations to logLR, which also depended on $\rho$ (using as a scale factor the inverse of $\frac{2\mu_g}{\sigma_g^2(1+\rho)}$; see *Figure 1b*). Because this conversion also applied to internal quantities, again we used a subjective estimate of the correlation to scale the bound, $\hat{\rho}_B$ (which we assumed could differ from $\hat{\rho}_{SD}$, as detailed below). Specifically, we scaled the bound as (see 'Materials and methods' for details):

$$B_\rho = B_0\frac{(1+\hat{\rho}_B)}{\sqrt{1+\rho}}\frac{1}{\sqrt{1+\hat{\rho}_{SD}}},$$

where $B_0$ is the bound height for the zero-correlation condition (which subsumes the components of the logLR scale factor that do not depend on $\rho$).

Consider two cases that illustrate how correlation misestimates could affect the decision process. First, an observer's internal encoding of the observation distribution could underestimate the correlation magnitude (i.e., $|\hat{\rho}_{SD}| < |\rho|$), but they then use this underestimate normatively to compute the weight of evidence. For this 'suboptimal encoding' case, $\hat{\rho}_B = \hat{\rho}_{SD} = \hat{\rho}$ and

$$B_\rho = B_0\frac{\sqrt{1+\hat{\rho}}}{\sqrt{1+\rho}}.$$

This formulation leads to the following predictions (*Figure 5b*):

- If $\hat{\rho} = \rho$, then $drift\,rate(\rho) = drift\,rate(0)$ and $B_\rho = B_0$: When correlations are perceived and estimated accurately, the drift rate and bound height are equal across correlations, giving equal average choices and RTs (e.g., *Figure 5b*, right-most column).
- If $\hat{\rho} < \rho$ and $\rho > 0$, then $drift\,rate(\rho) > drift\,rate(0)$ and $B_\rho < B_0$: When positive correlations are underestimated, the subjective evidence strength is greater than the objective evidence strength, corresponding to a higher drift rate and lower bound than when the correlations are estimated correctly. These effects cancel for the psychometric function (which depend only on the product of drift rate and bound height), leaving accuracy unchanged, but tend to produce faster RTs.

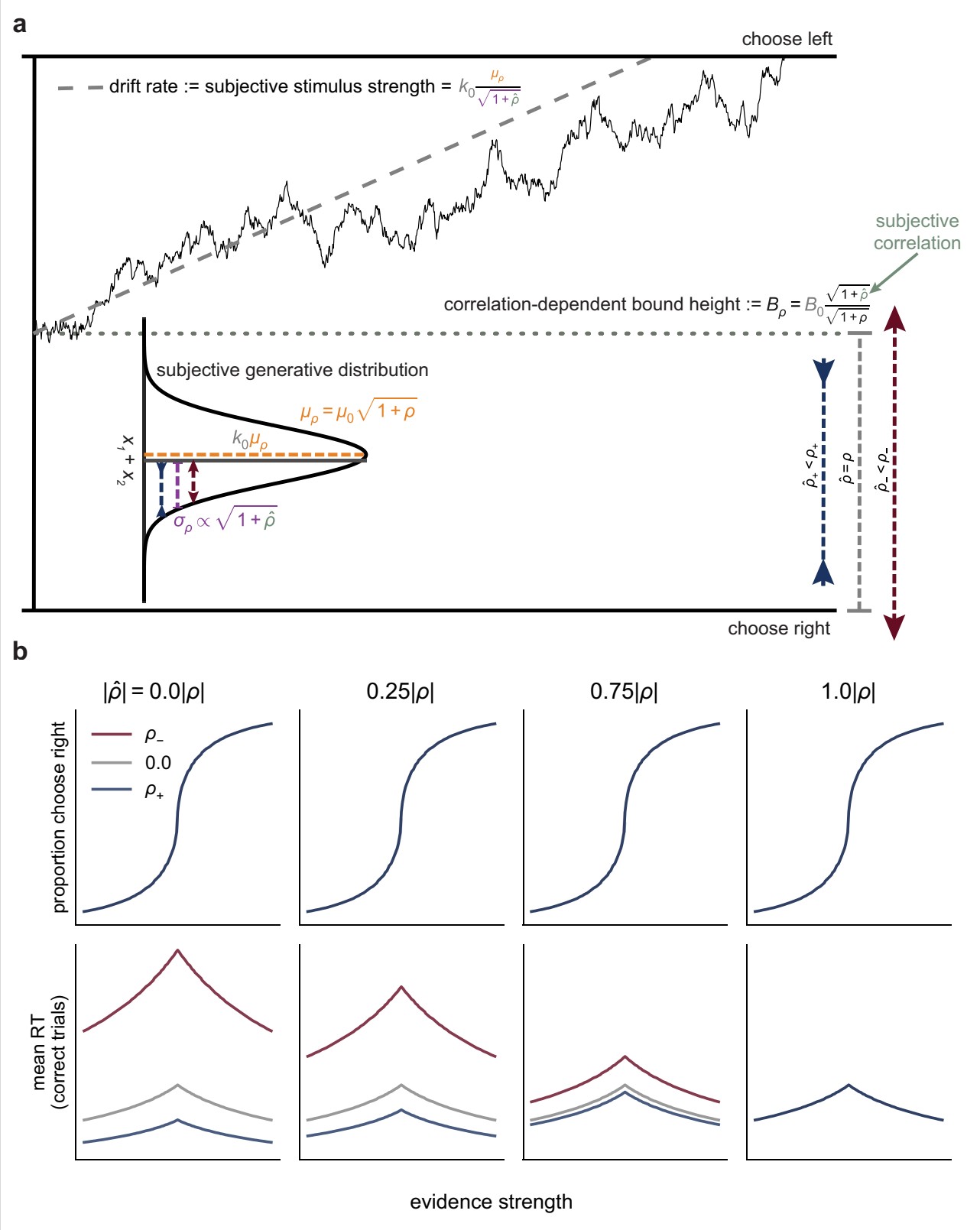

**Figure 5.** A drift-diffusion model (DDM) captures normative evidence weighing via bound-height adjustments. (**a**) In the DDM, sensory observations are modeled as samples from a Gaussian distribution (in the continuum limit). Evidence is accumulated over time as the decision variable until it reaches one of the two bounds, which terminates the decision in favor of the choice corresponding to that bound (here for simplicity we show fixed bounds, but in the fitting detailed below we use collapsing bounds). For pairs of correlated observations, altering the correlation between the pairs is equivalent to

*Figure 5 continued on next page*

*Figure 5 continued*

changing the standard deviation of the generative distribution of the sum of each pair, which affects the drift rate plus the scaling of the bound height (see 'Materials and methods'). We designed the task such that this scaling effect on drift rate was countered exactly by correlation-dependent changes in the mean of the generative distribution. Normative evidence weighing corresponds to correlation-dependent adjustments of the bound height that are functionally equivalent to scaling the observations to compute the true logLR. (b) Predictions from a DDM that implements normative bound-height adjustments but allows for subjective misestimates of the correlation. Colors correspond to three simulated correlation conditions (see legend and headings). Other parameters were chosen to approximate the fits to human data. Each column depicts predictions based on the same form of correlation-dependent bound scaling but with a different subjective correlation $\hat{\rho}$ (i.e., the correlation assumed by the observer), which was computed as a proportion of the objective correlation $\rho$ (computed on Fisher-z-transformed correlations that were then back-transformed). Given equal expected logLR across correlation conditions, underestimating the correlation $\left(\left|\hat{\rho}\right| < |\rho|\right.$, first three columns) leads to RT differences across the conditions, where the magnitude of the differences depends on the degree of underestimation (note that misestimating $\rho$ does not cause correlation-dependent changes in choice patterns predicted by the DDM because choice depends on the product of the drift and bound, in which the subjective terms cancel). Only $\hat{\rho} = \rho$ (rightmost column) produces exactly equal predicted RTs across conditions.

The online version of this article includes the following figure supplement(s) for figure 5:

**Figure supplement 1.** A drift-diffusion model (DDM) based on suboptimal evidence weighing.

---

- If $\hat{\rho} > \rho$ and $\rho < 0$, then *drift rate* $\left(\rho\right) < $ *drift rate* $\left(0\right)$ and $B_\rho > B_0$: When negative correlations are underestimated, the subjective evidence strength is less than the objective evidence strength, corresponding to a lower drift rate and higher bound than when the correlations are estimated correctly. These effects cancel for the psychometric function, leaving accuracy unchanged, but tend to produce slower RTs.

Second, an observer could encode the true $\rho$ but underestimate this correlation when computing the weight of evidence ($\hat{\rho}_B$). For this 'suboptimal weighing' case, $\hat{\rho}_{SD} = \rho$ and

$$B_\rho = B_0 \frac{\sqrt{1 + \hat{\rho}_B}}{\sqrt{1 + \rho}}.$$

Unlike suboptimal encoding, suboptimal weighing affects the speed-accuracy trade-off and thus predicts changes in both accuracy and RT (compare *Figure 5* with *Figure 5—figure supplement 1*).

Scaling the bound in these formulations follows conventions of the DDM, as detailed above, to facilitate interpretation of the parameters. These formulations also raise an apparent contradiction: the 'predefined' bound is scaled by subjective estimates of the correlation, but the correlation was randomized from trial to trial and thus could not be known in advance. However, scaling the bound in these ways is mathematically equivalent to using a fixed bound on each trial and scaling the observations to approximate logLR (see 'Materials and methods'). This equivalence implies that in the brain, effectively scaling a 'predefined' bound could occur when assigning a weight of evidence to the observations as they are presented.

## Human performance is consistent with correlation-dependent bound adjustments

Qualitatively, the participants' patterns of correlation-dependent RTs but correlation-independent choices were consistent with slight underestimates of the correlation (see *Figure 4*) resulting from suboptimal encoding (compare *Figures 3b and 5b*), not suboptimal weighing (compare *Figure 3b* and *Figure 5—figure supplement 1b*). To examine these effects more quantitatively, we fit six DDMs to each participant's data (see 'Materials and methods' for details). Prior to fitting these models, we confirmed that the DDM with a linear collapsing bound could generally account for choice and RT data from the zero-correlation condition (*Figure 6—figure supplement 1*). All models included five basic free parameters: one for the drift rate ($k_0$), two for the bound ($B_0$, $t_B$), one accounting for sensory and motor ('non-decision') processing times ($ndt$), and one for lapses ($\lambda$). The models differed in whether and how they accounted for changes in the correlation.

The six models we used were (1) a *base* model, which included no adjustments to evidence weighing based on the correlation; (2) a *drift* model, which included unconstrained, correlation-dependent adjustments in the drift rate, but not the bound (by fitting separate drift parameters for the negative, zero, and positive correlation conditions: $k_-$, $k_0$, $k_+$, respectively); (3) a *bound-$\hat{\rho}$* model, which included correlation-dependent adjustments to the bound based on subjective estimates of

the correlation (i.e., suboptimality in evidence weighing, with extra free parameters $\hat{\rho}_{B-}$ and $\hat{\rho}_{B+}$); (4) a *full-$\hat{\rho}$* model, which included normative evidence weighing that depended on subjective estimates of the correlation (i.e., suboptimality in encoding, where $\hat{\rho}_{B} = \hat{\rho}_{SD} = \hat{\rho}$, with extra free parameters $\hat{\rho}_{-}$ and $\rho_{+}$); (5) a *scaled-$\hat{\rho}$* model, which permitted suboptimality in both encoding and weighing (with extra free parameters $\hat{\rho}_{SD-}$, $\hat{\rho}_{SD+}$, $\hat{\rho}_{B-}$, and $\hat{\rho}_{B+}$); and (6) a *bound-$\hat{\rho}$ + drift* model, which was the same as the *bound-$\hat{\rho}$* model but also included unconstrained, correlation-dependent adjustments in the drift rate (with extra free parameters $k_{-}$, $k_{+}$, $\hat{\rho}_{B-}$, and $\hat{\rho}_{B+}$). For all models that did not explicitly fit $\hat{\rho}_{SD}$, we set $\hat{\rho}_{SD} = \rho$, which corresponds to the standard DDM assumption that subjective stimulus strength is related linearly to objective stimulus strength (*Palmer et al., 2005*).

Supporting our qualitative observations, these model fits indicated that the *full-$\hat{\rho}$* model best captured behavioral performance across correlation conditions (*Figure 6a*; see *Supplementary file 1* for average best-fitting parameters for all models). Thus, the participants' behavior was consistent with an accumulate-to-bound process that was scaled on each trial to form decisions based on a normative, correlation-dependent weighing of evidence that was encoded according to a slight misestimate of the correlation. This result was true for all correlation magnitudes individually, except for the 0.2 group, for which the fit statistics were equivocal between the *full-$\hat{\rho}$* and *bound-$\hat{\rho}$* models (*Figure 6b*; predictions of the two models are similar for low correlations and become more distinguishable as correlations increase; compare *Figure 6c* with *Figure 6—figure supplement 2*). The models that did not include correlation-dependent bound adjustments (the *base* and *drift* models) provided poor fits to the data (*Figure 6a*, *Figure 6—figure supplement 3*), underscoring the importance of changes in evidence weighing rather than simply changes in subjective stimulus strength for capturing the participants' performance.

## Human performance is consistent with a weight of evidence based on the approximated correlation

The *full-$\hat{\rho}$* model, which implements normative evidence weighing given a subjective estimate of the correlation, best accounted for behavioral differences across correlation conditions. We leveraged the fits to relate subjective correlation estimates to performance.

There was a strong relationship between the objective and subjective correlations used by each participant (*Figure 7a*; B=0.71, *t*(99)=47.92, p<0.001, Fisher *z*-transformed scale). This result confirms that the participants were sensitive to the correlations and used them to adjust their decision process. However, the slope of this relationship was less than one. That is, participants underestimated the objective correlation, on average (test of $\rho - \hat{\rho}$ (Fisher *z*-transformed scale): B=−0.18, *t*(199)=−13.29, p<0.001). This result is consistent with our hypothesis that their deviations from optimal behavior (i.e., unequal RTs across correlation conditions) resulted from subjective correlation estimates that were systematically lower than the true generative correlation. The fit (subjective) correlations also tended to be more variable for positive versus negative correlations (*Figure 7a*; mean value of the standard deviation of Fisher *z*-transformed estimates = 0.20 for positive correlation conditions, 0.09 for negative correlation conditions), possibly reflecting the weaker consequences of misestimating positive versus negative correlations on performance (*Figure 7b*). Despite these suboptimalities, overall performance was much closer to ideal (i.e., perfect encoding and use of the correlation) than if the participants ignored the correlations when computing the weight of evidence (*Figure 7b*).

We do not know how the participants formed their subjective estimates of the correlation or why in many cases these estimates tended to be biased toward zero. One possibility is that they formed an independent, empirical estimate on each trial, based on the observed samples. Consistent with this idea, these empirical correlations tended to be biased toward zero when based on a very limited number of samples, as expected (*Figure 7—figure supplement 1*; *Zimmerman et al., 2003*). These estimates might also have been biased toward their across-trial mean of zero, which could have served as a prior given that the estimates were based on limited data and thus were highly uncertain.

However, two lines of evidence argue against the idea that the participants computed an empirical estimate of the correlation on each trial. First, this idea predicts that the largest biases toward zero should occur when the estimates are based on the fewest number of samples. Contrary to this prediction, we found no reliable evidence that participants with the shortest average RTs (and thus who observed the fewest samples) tended to have subjective estimates that were most strongly biased towards zero (*Figure 7—figure supplement 2*). Second, there was no systematic relationship between

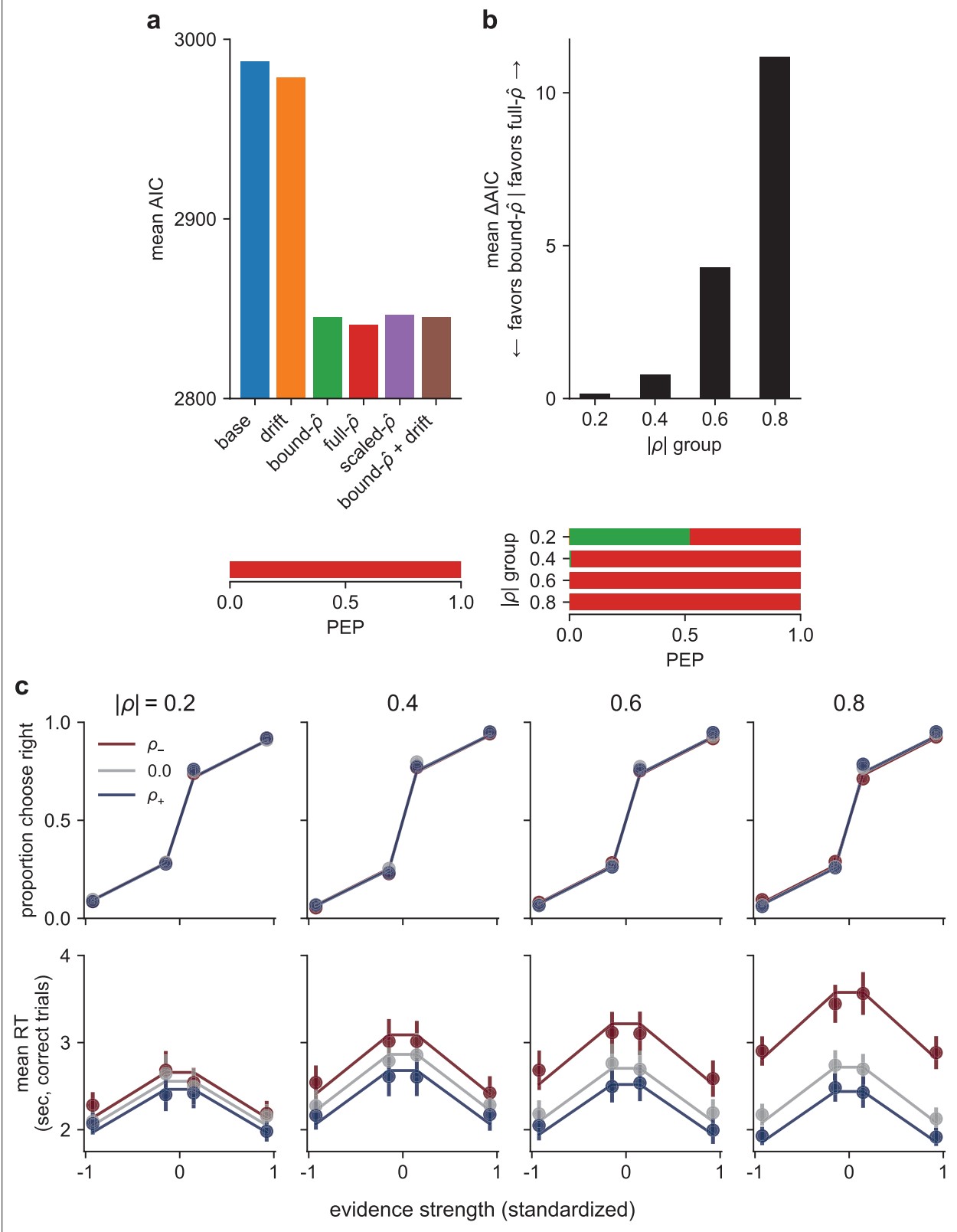

**Figure 6.** A drift-diffusion model (DDM) accounts for human behavior. (**a**) Model comparison: mean AIC (top) and protected exceedance probability (PEP; bottom), across all participants, for six different models, as labeled (see text for details). (**b**) Model comparison within each correlation-magnitude group, showing the difference in AIC between the *full-$\hat{\rho}$* and *bound-$\hat{\rho}$* models (top) and PEP over all models (bottom). Bar colors in the PEP plots correspond to the model colors in the top panel of (**a**). (**c**) Predictions from the *full-$\hat{\rho}$* DDM (lines) plotted against participant data (points) for choice (top)

*Figure 6 continued on next page*

*Figure 6 continued*

and response time (RT) (bottom) for each correlation-magnitude group (columns, labels at top). Predictions and data are averaged across participants. Colors correspond to the three correlation conditions (see legend). Error bars are SEM.

The online version of this article includes the following figure supplement(s) for figure 6:

**Figure supplement 1.** A collapsing-bound model accounts for behavior.

**Figure supplement 2.** Predictions from the *bound-$\hat{\rho}$* DDM (lines) plotted against participant data (points) for choice (top) and response time (RT) (bottom) for each correlation-magnitude group (columns, labels at top).

**Figure supplement 3.** Predictions (lines) from the *base* model (**a**) and the *drift* model (**b**) plotted against participant data (points) for choice (top) and response time (RT) (bottom) for each correlation-magnitude group (columns, labels at top).

the best-fitting $\hat{\rho}$ and the mean empirical $\rho$ computed per participant for each correlation condition (*Figure 7—figure supplement 3*). These results suggest that the participants did not compute the correlation explicitly on each trial and thus instead might have used pattern matching (e.g., quickly assessing if the first few samples seem consistent with other observations from the negative-, zero-, or positive-correlation stimuli they already observed) or other heuristics to make correlation-dependent adjustments to the decision process.

These correlation-dependent differences in the decision process also did not seem to reflect ongoing adjustments that might involve, for example, feedback-driven learning specific to this task. In particular, the participants tended to exhibit some learning over the course of the task, involving substantial decreases in RT (the mean ± SEM difference in RT between the first and second half of the task, measured across participants, was 0.71±0.06 s, respectively, Mann–Whitney test for $H_0$: median difference = 0, p<0.001) at the expense of only slight decreases in accuracy (0.02 ± 0.00% correct, p=0.004). These trends reflected a tendency to use slightly higher drift rates (*Figure 8a*) and lower decision bounds (*Figure 8b*) in the latter half of the task, a pattern of results that is consistent with previous reports of practice effects for simple decisions (*Balci et al., 2011*; *Dutilh et al., 2009*). However, these adjustments were not accompanied by similar, systematic adjustments in the participants' subjective correlation estimates, which were similar in the first versus second half of the task (*Figure 8c*). This conclusion was supported by a complementary analysis showing that linear changes in RT as a function of trial number within a session tended to be the same for positive- and negative-correlation trials, as expected for stable relationships between correlation and RT (Wilcoxon rank-sum test for $H_0$: median difference in slope = 0, p<0.05 for just one of eight evidence strength × correlation magnitude conditions, after accounting for multiple comparisons via Bonferroni correction). Thus, participants' decisions appeared to be based on relatively stable estimates of the stimulus correlations that could be determined and used effectively on a trial-by-trial basis.

## Discussion

This preregistered study addressed a fundamental question in perceptual decision-making: how do people convert sensory observations into a weight of evidence that can be used to form a decision about those observations? This question is important because evidence weighing affects how information is combined and accumulated over multiple sources and over time, ultimately governing the speed and accuracy of the decision process (*Bogacz et al., 2006*; *Wald and Wolfowitz, 1948*). To answer this question, we focused on correlations between observations, which are common in the real world, often ignored in laboratory studies, and can have a dramatic impact on the amount of evidence provided by a given set of observations. For simple perceptual decisions with correlated observations, the normative weight of evidence that accounts for these correlations can be expressed as a logLR. We showed that human participants make decisions that are approximately consistent with using this normative quantity, mitigating changes in decision speed and/or accuracy that would result from ignoring correlations. Below we discuss the implications of these findings for our understanding of the computations and mechanisms the brain uses to form simple decisions.

Previous support for the idea that human decision-makers can weigh and combine multiple pieces of evidence following normative principles has come from two influential lines of research. The first is studies of perceptual cue combination. Perceptual reports based on cues from multiple sensory modalities, or multiple cues from the same modality, often reflect weights of evidence that scale with

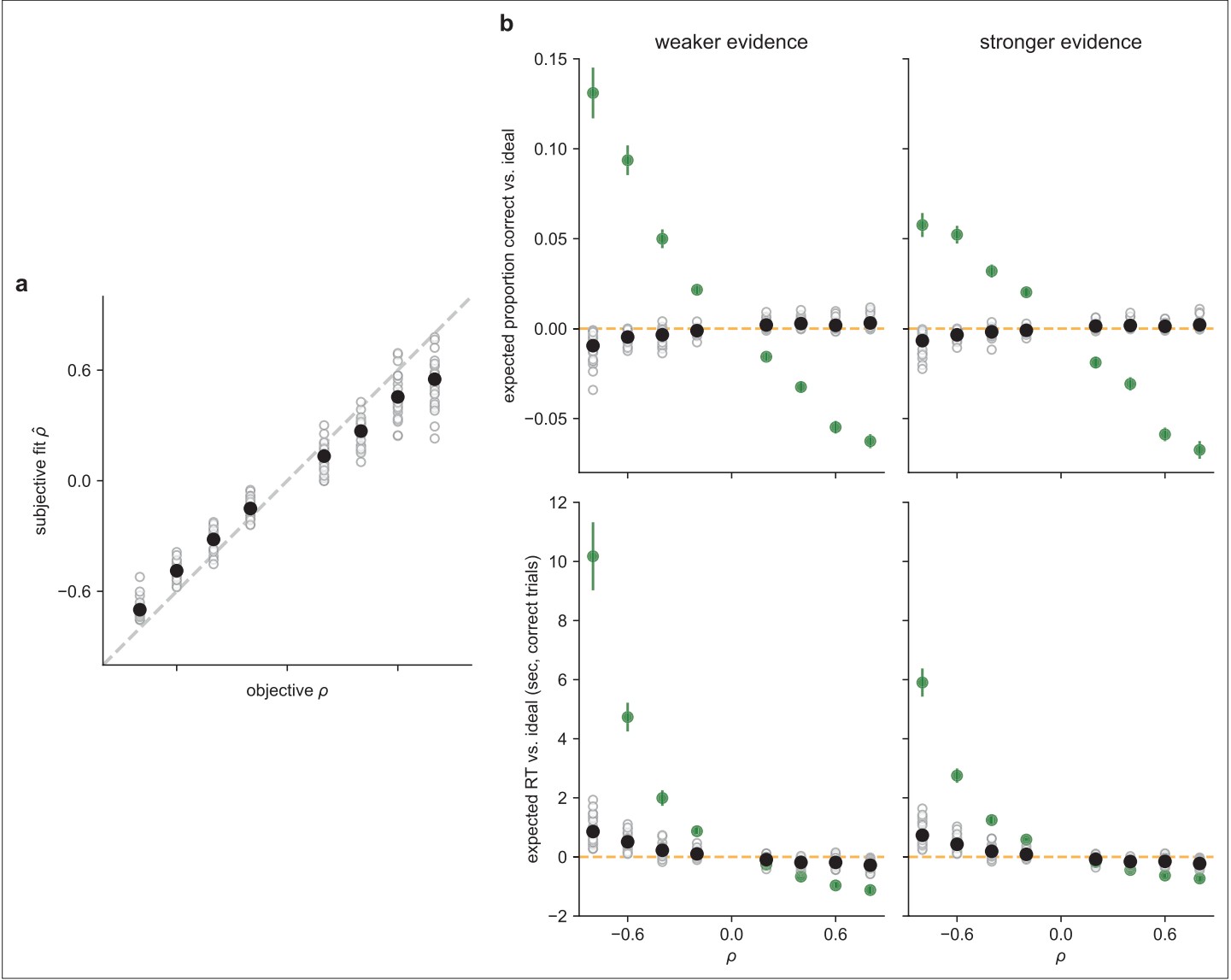

**Figure 7.** Participants used near-optimal correlation estimates, with slight biases away from extreme values. (**a**) The subjective fit correlation ($\hat{\rho}$) from the DDM as a function of the objective correlation ($\rho$). Open circles are the fits from individual participants. Closed circles are the means per correlation condition (means were computed on Fisher $z$-transformed values and then back-transformed). Error bars (not visible in most cases) are SEM. The dashed line is the unity line. (**b**) Expected accuracy (top) and RT (bottom) from the DDM fits to data from each participant (open circles) for weak (left) or strong (right) evidence, relative to an ideal observer (orange line) simulated with the per-participant DDMs using the true, objective correlation ($\hat{\rho} = \rho$). Black circles are mean values across participants. Green circles are data simulated with the per-participant DDMs for a naïve observer that used the same subjective fit correlation but did not use the correlation to normatively adjust the bound (i.e., $\hat{\rho}_{SD} = \hat{\rho}$, $\hat{\rho}_B = 0$). Note that predicted performance that appears slightly better than ideal for positive correlations is an artifact of varying the correlation independently of the drift rate in our simulations (for illustrative purposes), when they would both presumably be affected by suboptimalities in encoding.

The online version of this article includes the following source data, source code, and figure supplement(s) for figure 7:

**Figure supplement 1.** Empirical correlation estimates.

**Figure supplement 2.** Subjective correlation (best-fitting $\hat{\rho}$) versus mean response time (RT) for each participant.

**Figure supplement 2—source data 1.** Best-fitting parameter values used to create the figures generated by *Figure 7—figure supplement 2—source code 1* and *Figure 7—figure supplement 3—source code 1*.

**Figure supplement 2—source code 1.** Code to generate *Figure 7—figure supplement 2*.

**Figure supplement 3.** No systematic relationship between the mean of the empirical correlation computed on each trial (ordinate) and the best-fitting subjective correlation (abscissa) for positive (squares) and negative (diamonds) correlations.

**Figure supplement 3—source data 1.** Average observed correlation per subject and condition used in *Figure 7—figure supplement 3—source*

*Figure 7 continued on next page*

*Figure 7 continued*

**code 1**.

**Figure supplement 3—source code 1.** Code to generate *Figure 7—figure supplement 3*.

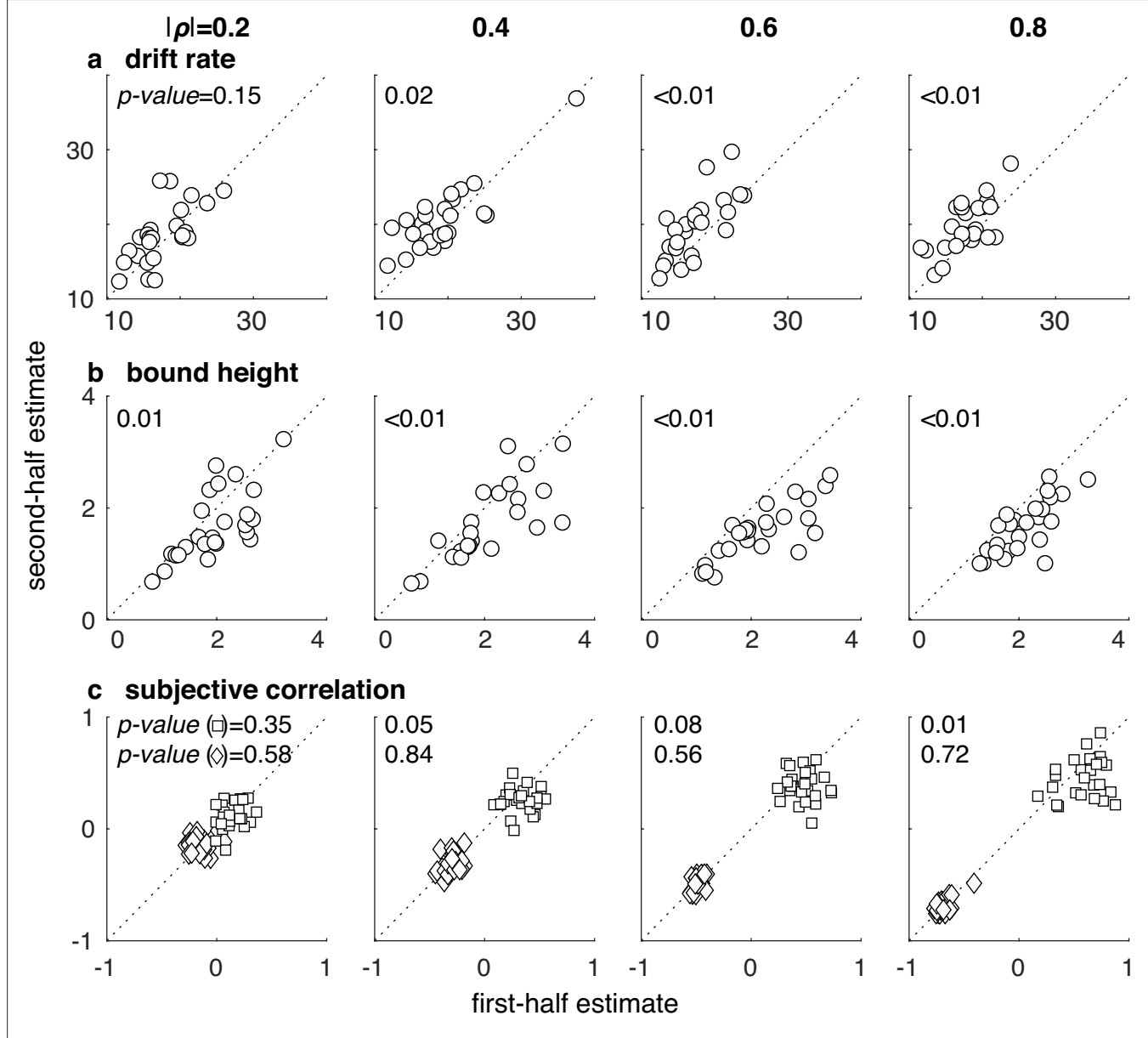

**Figure 8.** Participants used stable estimates of the correlations, even as they adjusted other components of the decision process over the course of a session. Each panel shows a scatterplot of DDM parameters estimated using the first (abscissa) versus second (ordinate) half of trials from a given participant. Points are data from individual participants. Columns are correlation-magnitude group, and rows are (**a**) drift rate, $k_0$; (**b**) bound height, $B_0$; (**c**) estimates of positive ($\hat{\rho}_+$ squares) and negative ($\hat{\rho}_-$; diamonds) subjective correlations. p-Values are for a Wilcoxon rank-sum test for $H_0$: median difference between the first- and second-half parameter estimates across participants = 0, uncorrected for multiple comparisons (only effects labeled as p<0.01 survived Bonferroni correction).

The online version of this article includes the following source data for figure 8:

**Source code 1.** Code to generate *Figure 8*.

**Source data 1.** Best-fitting parameter values for each half of the task used in *Figure 8—source code 1*.

the relative reliability of each cue, consistent with Bayesian theory (*Ernst, 2005*; *Noppeney, 2021*). The second is studies of evidence accumulation over time. The relationship between speed and accuracy for many decisions can be captured by models like the DDM that assume that the underlying decision process involves accumulating quantities that are often assumed to be (scaled) versions of the logLR (*Bogacz et al., 2006*; *Edwards, 1965*; *Gold and Shadlen, 2001*; *Laming, 1968*; *Stone, 1960*).

Central to the interpretation of these studies, and ours, is understanding the scale factors that govern evidence weights. In their simplest forms, these scale factors are scalar values that are multiplied by the observed stimulus strength to obtain the weight of evidence associated with each observation. These weights are then combined (e.g., by adding them together if they are in the form of logLR) to form a single decision variable, which is then compared to one or more criterion values (the bounds) to arrive at a final choice, as in the DDM (see *Figure 5a*). Thus, as long as there is a linear relationship between subjective stimulus strength and logLR, then using an appropriate, multiplicative scale factor to compute the weight of evidence (either scaling the observations or the bound, depending on the particular algorithmic implementation) can support normative decision-making.

These kinds of decision processes have been studied under a variety of conditions that have provided insights into how the brain scales observations to arrive at a weight of evidence. In the simplest evidence-accumulation paradigms, objective stimulus strength is held constant across decisions within a block. In this case, the same scale factor can be applied to each decision within a block, and normative changes in scaling across blocks are equivalent to shifting the decision bound to account for changes in objective stimulus strength. Results from studies using these paradigms have been mixed, including bounds that do (*Malhotra et al., 2017*; *Starns and Ratcliff, 2012*) or do not (*Balci et al., 2011*) vary across blocks. Interpretation of these studies is complicated by the fact that the participants are typically assumed to have the goal of maximizing reward rate, which is a complicated function of multiple task parameters, including stimulus strength and timing (*Bogacz et al., 2006*; *Zacksenhouse et al., 2010*). Under such conditions, failure to take stimulus strength into account, or failure to do so optimally, could be a result of the particular strategy adopted by the decision-maker rather than a failure to accurately estimate the appropriate scale factor. For example, several studies found that people deviate from optimal to a greater degree in low-stimulus-strength conditions because they value accuracy and not solely reward rate (*Balci et al., 2011*; *Bohil and Maddox, 2003*; *Starns and Ratcliff, 2012*). Additionally, deviations from optimal bounds can depend on the uncertainty with which task timing is estimated, rather than uncertainty in estimates of stimulus strength (*Zacksenhouse et al., 2010*), suggesting that people can estimate stimulus strength even if they do not use it as prescribed by reward-rate maximization. By fixing expected logLR (objective evidence strength) across conditions, we avoided many of these potential confounds and isolated the effects of correlations on behavior.

Objective stimulus strength is also often varied from trial-to-trial in evidence-accumulation tasks. Under these conditions, the standard SPRT, and the DDM as its continuous-time equivalent (*Bogacz et al., 2006*), are no longer optimal (*Deneve, 2012*; *Drugowitsch et al., 2012*; *Moran, 2015*). These models typically assume that the same scale factor is used on each trial, but different scale factors are needed to compute the normative weight of evidence (logLR) for different stimulus strengths. These considerations have led some to argue that it is highly unlikely that humans perform optimal computations, particularly under conditions of heterogenous stimulus strengths, because the precise stimulus statistics needed to compute the logLR are assumed to be unavailable or poorly estimated (*Ratcliff et al., 2016*; *Ratcliff and McKoon, 2008*). Relatedly, if decision-makers set their bounds according to the true logLR for each stimulus (equivalent to the goal of maintaining, on average, the same level of accuracy across stimuli), the psychometric function should be flat as a function of stimulus strength, whereas RTs should decrease with increasing stimulus strength. That decisions are both more accurate and faster with increasing stimulus strength argues strongly against the idea that people set bounds based on a fixed expected accuracy or that the accumulated evidence is scaled exactly proportional to the logLR (*Hanks et al., 2011*).

However, several modeling and empirical studies have shown that it is possible to adjust how decisions are formed about stimuli whose statistics vary from trial to trial, in a manner that is consistent with trying to use optimal (or near-optimal) forms of the weight of evidence. These adjustments include scaling the decision variable and/or decision bounds within a trial according to online

estimates of stimulus strength or some proxy thereof, particularly when the distribution of stimulus-strength levels is known (*Deneve, 2012*; *Drugowitsch et al., 2012*; *Hanks et al., 2011*; *Huang and Rao, 2013*; *Malhotra et al., 2018*; *Moran, 2015*). One possible proxy for stimulus strength is the time elapsed within a trial (*Drugowitsch et al., 2012*; *Hanks et al., 2011*; *Kiani and Shadlen, 2009*; *Malhotra et al., 2018*): the more time has passed in a trial without reaching a decision bound, the more likely that the evidence is weak. Under certain conditions, human decision-making behavior is consistent with making such adjustments, for example by using decision bounds that 'collapse' over time (*Drugowitsch et al., 2012*; *Malhotra et al., 2017*; *Palestro et al., 2018*).

Our results imply that outside relatively simple cases involving statistically independent observations, elapsed time cannot serve as a sole proxy for stimulus strength. In particular, correlations between pairs of observations can complicate the relationship between the strength of evidence provided by individual observations and elapsed time. For example, for our task negative correlations lead to slower decisions than positive correlations if the observations are treated as uncorrelated, when in fact the objective evidence strength is stronger for negative correlations than positive correlations. Therefore, in more general settings elapsed time should be combined with other relevant statistics, such as the correlation, to determine an appropriate weight of evidence. In support of this idea, our data are consistent with decisions that used collapsing bounds, which helped adjust the decision process when the objective evidence strength changed within each correlation condition (i.e., the low and high values of evidence strength that corresponded to substantial modulations of the psychometric and chronometric functions). However, collapsing bounds alone, without additional correlation-dependent computations, could not account for our behavioral results, which by design used values of objective evidence strength that did not change across correlation conditions.

Our results are in stark contrast with the literature on correlations in behavioral economics, which suggests that people fail to use correlations appropriately to inform their decision-making. For example, when combining information from multiple sources (e.g., for financial forecasts: *Budescu and Yu, 2007*; *Enke and Zimmermann, 2017*; *Hossain and Okui, 2021*; *Maines, 1996*; *Maines, 1990*; or constructing portfolios of correlated assets: *Eyster and Weizsacker, 2016*; *Laudenbach et al., 2023*), most participants exhibit 'correlation neglect' (i.e., partially or fully failing to account for correlations), which often leads to reduced decision accuracy. Positive correlations have also been proposed to lead to overconfidence, which has been attributed to failing to account for redundancy (*Eyster and Rabin, 2010*; *Glaeser and Sunstein, 2009*; *Ortoleva and Snowberg, 2015*) or to the false assumption that consistency among information sources suggests higher reliability (*Kahneman and Tversky, 1973*).

These discrepant results are likely a result of the vast differences in task designs between those studies and ours. Those tasks tended to present numerical stimuli representing either small samples of correlated sources or explicitly defined correlation coefficients, often in complicated scenarios. Under such conditions, participants may fail to recognize the correlation and its importance or they may not be statistically sophisticated enough to adjust for it even if they do (*Enke and Zimmermann, 2017*; *Maines, 1996*). In contrast, highly simplified task structures increase the ability to account for correlations (*Enke and Zimmermann, 2017*). Nevertheless, even in simplified cases, decisions in descriptive scenarios likely rely on very different cognitive mechanisms than decisions that, like ours, are based directly on relatively simple sensory stimuli. For example, decisions under risk can vary substantially when based on description versus direct experience (*Hertwig and Erev, 2009*), and giving passive exposure to samples from distributions underlying two correlated assets can alleviate correlation neglect in subsequent allocation decisions (*Laudenbach et al., 2023*).

These differences likely extend to how and where in the brain correlations are represented and used (or not) to inform different kinds of decisions. For certain perceptual decisions, early sensory areas may play critical roles. For example, when combining multiple visual cues to estimate slant, some observers' estimates are consistent with assuming a correlation between cues, which is sensible because the cues derive from the same retinal image and likely overlapping populations of neurons (*Oruç et al., 2003*; *Rosas et al., 2007*). The combination of within-modality cues is thought to be encapsulated within the visual system, such that observers have no conscious access to the individual cues (*Girshick and Banks, 2009*; *Hillis et al., 2002*). These results suggest that the visual system may have specialized mechanisms for computing correlations among visual stimuli (which may or may not involve the well-studied, but different, phenomena of correlations in the patterns of firing rates of

individual neurons; *Cohen and Kohn, 2011*) that are different than those used to support higher-order cognition. Given our finding that suboptimal evidence weighing reflected subjective misestimates of the correlation at the level of encoding, these putative encoding mechanisms appear to be tightly coupled to those that convert observations into a weight of evidence.

The impact of correlations on the weight of evidence depends ultimately on the type of correlation and its relationship to other statistical features of the task environment and to intrinsic correlations in the brain (*Averbeck and Lee, 2006*; *Bhardwaj et al., 2015*; *Hossain and Okui, 2021*; *Hu et al., 2014*; *Moreno-Bote et al., 2014*). We showed that this impact can be substantial for a particular form of correlation, and that human decision-makers' sensitivity to correlations does not seem to require extensive, task-specific learning and can be adjusted flexibly from one decision to the next. Further work that pairs careful manipulation of task statistics with neural measurements could provide insight into how the brain tracks stimulus correlations and computes the weight of the evidence to support effective decision-making behaviors under different conditions.

## Materials and methods

### Participants

One hundred human participants took part in this online study (42 males, 43 females, 3 others, 12 N/A; median age: 24 years, range 18–53, 1 N/A), each of whom provided informed consent via button press. Human protocols were approved and determined to be Exempt by the University of Pennsylvania Internal Review Board (IRB protocol 844474). Participants were recruited using the Prolific platform (https://www.prolific.com/). They were paid a base amount of $9.00 for a projected completion time of 1 hour. They also could receive a bonus of up to $8, depending on task performance (see below).

### Behavioral task

The task was developed in PsychoPy (v. 2021.1.4; *Peirce, 2019*), converted to JavaScript (PyschoJS), and run on the online experiment hosting service Pavlovia (https://pavlovia.org/), via functionality integrated into PsychoPy. On each trial, the participant saw a sequence of observations. Each observation consisted of two stars displayed simultaneously. The stars' horizontal positions were generated from a bivariate Gaussian distribution with equal means and variances for each star position and a correlation between star positions that changed from trial-trial-to-trial (the generative distribution), while their vertical position was fixed in the center of the display. The stars were generated by either a 'left' source or a 'right' source, chosen randomly with equal probability on each trial. The two sources were equidistant from the vertical midline of the screen, corresponding to equal means of the generative distribution with opposite signs. To prevent stars from being drawn past the edge of the display, their positions were truncated to a maximum value of 0.7, in units of relative window height. For a standard 16:9 monitor at full screen, this procedure implies that positions could not take on values past 78.8% of the distance from the center of the screen to the edge. Within a trial, new observations were generated from the underlying source distribution every 0.2 s. Participants were instructed to indicate whether the stars were being generated by the left or the right source once they believed they had accumulated enough noisy information to make an accurate decision.

Each participant was assigned randomly to one of the four correlation-magnitude groups ($|\rho|$=0.2, 0.4, 0.6, or 0.8; 25 participants per group) and completed 768 trials, which were divided into 4 blocks of 192 trials each, with brief breaks between blocks. Within each block, there were 12 different stimulus conditions varied pseudo-randomly from trial-to-trial, per participant: 2 sources (left, right) × 2 evidence strengths (low, high) × 3 correlations ($\rho_-$, 0.0, $\rho_+$). Within each block, the trials were divided into 16 sets, with one trial of each condition per set. Each condition was presented in random order within a set, such that all 12 conditions were presented once before the next repetition of a given condition, resulting in 64 total repetitions of each condition across the experiment. Participants received 1 point for each correct choice and −2 points for each incorrect choice (their total points could never go below zero). The total number of points received by the end of the task was divided by the total possible points (768), and that proportion of $8 was awarded as the bonus.

Prior to completing the main task, each participant completed first a set of training trials, then a staircase procedure to standardize task difficulty across participants. We used a 3-down, 1-up staircase procedure to identify each participant-specific evidence-strength threshold (i.e., by varying the mean

of the star-generating distribution while holding its standard deviation at a constant value of 0.1, in units of relative window height) that resulted in a target accuracy of 79.4% in the zero-correlation condition (*García-Pérez, 1998*). Staircase trials were presented at a fixed-duration of 1.4 s, which in pilot data was roughly the mean RT in the free-response paradigm used in the main task, to equate the amount of information provided to each participant and avoid potential individual differences in the speed-accuracy trade-off. The high and low evidence-strength conditions used in the main task were then defined as 0.4 and 2.5 times each participant's evidence-strength threshold, respectively.

Because the staircase procedure should standardize accuracy across participants, performance that is much lower than the target accuracy can be interpreted as a failure of the staircase procedure, a failure of the participant to maintain engagement in the task, or both. Therefore, we kept recruiting participants until we had 25 in each correlation-magnitude group with task performance at 70% or higher. No more than three candidate participants in each group were excluded based on this criterion.

## Ideal-observer analysis

Bivariate-Gaussian observations $(x_1, x_2)$ with equal means $\mu_g$, standard deviations $\sigma_g$, and correlation $\rho$ are distributed as

$$p\left(x_1, x_2 | S\right) = \frac{1}{2\pi\sigma_g^2\sqrt{1-\rho^2}} exp\left(-\frac{1}{2\left(1-\rho^2\right)}\left[\left(\frac{x_1-\mu_g}{\sigma_g}\right)^2 + \left(\frac{x_2-\mu_g}{\sigma_g}\right)^2 - 2\rho\frac{\left(x_1-\mu_g\right)\left(x_2-\mu_g\right)}{\sigma_g^2}\right]\right),$$

where $S$ is the generative source. For the problem of choosing between two such generative sources, $S_0$ and $S_1$, the normative weight of evidence can be computed using the log-likelihood ratio, $\log LR = \log\left(\frac{p\left(x_1, x_2 | S_1\right)}{p\left(x_1, x_2 | S_0\right)}\right)$. For sources with means $\mu_0$ and $\mu_1$ and equal $\sigma_g$ and $\rho$, the logLR reduces to

$$logLR_{S_1,S_0}(x_1, x_2) = \frac{\left(\mu_1 - \mu_0\right)}{\sigma_g^2\left(1+\rho\right)}\left[\left(x_1 + x_2\right) - \left(\mu_1 + \mu_0\right)\right].$$

Our task had equal and opposite generative means, $\mu_0 = -\mu_1 = \mu_g$. Under these conditions, the logLR further simplifies to

$$logLR_{S_1,S_0}(x_1, x_2) = \frac{2\mu_g}{\sigma_g^2\left(1+\rho\right)}\left(x_1 + x_2\right).$$

This logLR is a weight of evidence composed of the sum of the observations (for our task corresponding to the horizontal locations of the two stars) multiplied by a scale factor that depends on the generative properties of the sources. Because this logLR, which is expressed in terms of bivariate observations $(x_1, x_2)$, depends only on the sum of the observations, it is equivalent to a logLR expressed in terms of univariate observations composed of the sum of each pair (i.e., $logLR_{S_1,S_0}\left(x_1 + x_2\right)$, where $(x_1 + x_2) \sim N(2\mu_g, \sqrt{2\sigma_g^2(1+\rho)})$; see *Figure 2c*). The logLR for a sequence of these (identically distributed, paired) observations is the sum of the logLRs for the individual (paired) observations.

We defined the objective evidence strength for a given condition as the expected value of the logLR for a single (paired) observation:

$$\text{objective (expected) evidence strength}:= E[logLR_{S_1,S_0}\left(x_1, x_2\right)] = \frac{4\mu_g^2}{\sigma_g^2\left(1+\rho\right)}.$$

Therefore, to equate the evidence strength between two conditions with equal $\sigma_g$, but one with correlation $= 0$ and one with correlation $\rho \neq 0$, we adjusted the generative mean of condition $\rho$ to offset the correlation-dependent scale factor $\frac{1}{1+\rho}$:

$$E[logLR_\rho] = E[logLR_0]$$

$$\frac{4\mu_\rho^2}{\sigma_g^2(1+\rho)} = \frac{4\mu_0^2}{\sigma_g^2}$$

$$\mu_\rho = \mu_0\sqrt{1+\rho}.$$

## Drift-diffusion modeling

In the DDM, noisy evidence is accumulated into a decision variable until reaching one of the two bounds, representing commitment to one of two choices (e.g., left or right). In general, the average rate of accumulation is governed by the drift rate:

$$drift\ rate = k\frac{\mu_g}{\sigma_g},$$

where $\mu_g$ and $\sigma_g$ are the mean and standard deviation of the generative distribution of the observations (which, as detailed above, for our task can be expressed as the distribution of sums of pairwise observation, $x_1 + x_2$). The drift parameter $k$ captures subjective scaling of the objective stimulus strength (i.e., the SNR, $\frac{\mu_g}{\sigma_g}$), which accounts for individual differences in perceptual sensitivity and other factors.

There is an arbitrary degree of freedom in these and related models, which form equivalence classes when the decision variable and decision bound are both scaled in the same way (***Green and Swets, 1966***; ***Palmer et al., 2005***). Fixing this extra degree of freedom in the DDM is typically accomplished by setting $\sigma_g = 1$, which causes the drift rate and bound height to be scaled implicitly by the standard deviation of the observation distribution (***Palmer et al., 2005***). This formulation is straightforward when stimulus strength is varied via changes in only signal, $\mu_g$, and not noise, $\sigma_g$. In that case, the scaling is constant across signal strengths and thus typically simply assumed to be captured by the drift and bound parameters.

However, our task included correlation-dependent effects on both signal and noise. To specify the stimulus strength in the model when noise varies, both the drift-rate and bound-height terms must be scaled by the noise, which we implemented by scaling both terms by the correlation-dependent component of the noise (i.e., the standard deviation, or SD, of the generative process), $\sqrt{1 + \hat{\rho}_{SD}}$, . Here $\hat{\rho}_{SD}$ is a fit parameter corresponding to the subjective correlation experienced by the observer, which can deviate from the objective correlation. Unlike subjective scaling of signal strength, subjective deviations in the noise cannot be assumed to be absorbed by the drift-rate (i.e., $k$) or bound-height parameters because of the nonlinear effects of the noise. Note also that this formulation assumes that $\sigma_g$, the generative standard deviation in the zero-correlation condition, is absorbed into the drift-rate and bound-height parameters, which serve as an important baseline for the correlation-based adjustments. Therefore, the drift rate in our model was formulated as

$$drift\ rate(\rho) = k_0 \frac{\mu_\rho}{\sqrt{1 + \hat{\rho}_{SD}}} = \frac{k_0}{\sqrt{1 + \hat{\rho}_{SD}}} \mu_0 \sqrt{1 + \rho},$$

where $k_0$ is a fit parameter corresponding to the drift rate in the zero-correlation condition, $\mu_0$ is the true (objective) generative mean in the zero-correlation condition, and $\sqrt{1 + \rho}$ is the scale factor we used to scale the generative mean to equate objective evidence strength across correlation conditions.

To implement normative evidence weighing in the DDM, we started by assuming that the evidence distribution was based on the logLR estimated for each observed sample pair (***Figure 2c***). Converting the observation distribution to the evidence distribution entails scaling the observations by the signal and noise characteristics of the observation distribution, as derived above. In the DDM, this scaling of the evidence is equivalent to (1) assuming that the decision variable accumulates momentary evidence of the form $(x_1 + x_2)$, and then (2) dividing the bound height by the appropriate scale factor. An alternative approach would be to scale both the signal and noise components of the DDM by the scale factor. However, scaling the bound is simpler and maintains the conventional interpretation of the DDM parameters in which the bound reflects the decision-related components of the evidence accumulation process, and the drift rate represents sensory-related components. Accordingly, we scaled the bound height as

$$B_\rho = B\frac{\sigma_g^2 (1 + \hat{\rho}_B)}{2\mu_0\sqrt{1 + \rho}} \frac{1}{\sqrt{1 + \hat{\rho}_{SD}}} = B_0 \frac{(1 + \hat{\rho}_B)}{\sqrt{1 + \rho}} \frac{1}{\sqrt{1 + \hat{\rho}_{SD}}}.$$

where the first scale factor is the reciprocal of the normative evidence-weighing term that converts observations to logLR, as derived above; the second scale factor is the scaling by the correlation-dependent component of the noise; and $B_0$ is a fit parameter corresponding to the bound height in the zero-correlation condition (that absorbs the constant terms $\frac{\sigma_g^2}{2\mu_0}$). We include a subjective fit

correlation $\hat{\rho}_B$ in the correlation-dependent evidence-weighing term that was different than the one we used in the noise term ($\hat{\rho}_{SD}$), because in principle an observer may fail to appropriately scale the evidence by the observed correlation (e.g., a naïve observer who ignores the correlation will have $\hat{\rho}_B = 0$), even though the correlation can still influence performance through its effect on the noise.

We focus on two special cases of this model. When $\hat{\rho}_B = \hat{\rho}_{SD} = \hat{\rho}$,

$$B_\rho = B_0 \frac{\sqrt{1 + \hat{\rho}}}{\sqrt{1 + \rho}}.$$

We refer to this as the '*full-$\hat{\rho}$*' model because the subjective correlation estimate $\hat{\rho}$ is the same in both the drift and bound terms. This formulation is equivalent to assuming that misestimates of correlation magnitude are reflected in how the brain converts the observations to evidence, such that the correlation reflected in the internal observation distribution (which in the model is used to scale both the drift rate and bound height) is also used to compute logLR (which in the model is used to scale the bound height). This formulation predicts that misestimated correlations (i.e., $\hat{\rho} \neq \rho$) affect the chronometric function (because changes in the subjective scaling of the drift rate and bound height result in correlation-dependent differences in the evidence per time step, and thus the time taken to reach the bound) but not the psychometric function (which depends only on the product of the drift rate and bound height, and thus the $\rho$- and $\hat{\rho}$-dependent terms cancel and accuracy is constant across correlations; *Palmer et al., 2005*; also see *Figure 5*).

In the second case, when $\hat{\rho}_{SD} = \rho$,

$$B_\rho = B_0 \frac{\sqrt{1 + \hat{\rho}_B}}{\sqrt{1 + \rho}}.$$

This '*bound-$\hat{\rho}$*' model assumed that the statistics of the internal observation distribution followed objective $\rho$, and only the conversion of the observations to a weight of evidence was based on subjective $\hat{\rho}_B$ (i.e., the drift rate simplifies to $k_0\mu_0$, and the bound height was set as $B_\rho$, defined above). This formulation predicts that misestimated correlations affect both the chronometric and psychometric function (because the mis-specified weight of evidence results in correlation-dependent differences in the bound height and thus changes in the speed-accuracy trade-off; see *Figure 5—figure supplement 1*).

We tested four additional variants of the DDM. The *base* model assumed no correlation-dependent scaling of the bound height. The *drift* model allowed for unconstrained correlation-dependent changes to the drift rate via three drift parameters fit separately for the three correlation conditions: $k_-, k_0, k_+$. The *scaled-$\hat{\rho}$* model fit $\hat{\rho}_{SD}$ and $\hat{\rho}_B$ separately, allowing subjective evidence weighing to diverge from the observed correlation. The *bound-$\hat{\rho}$ + drift* model used the form of the bound from the *bound-$\hat{\rho}$* model and included unconstrained, correlation-dependent adjustments in the drift rate as in the *drift* model.

For all models that fit subjective correlation estimates, separate parameters were used for the positive and negative correlation conditions ($\hat{\rho}_+$, $\hat{\rho}_-$). Additionally, for all models that did not fit $\hat{\rho}_{SD}$, we set $\hat{\rho}_{SD} = \rho$, which is equivalent to assuming that the objective, correlation-dependent stimulus strength was encoded correctly (but then could also be scaled linearly by $k_0$, as in the standard DDM; *Palmer et al., 2005*). All models also included a non-decision time, *ndt*, that captures the contributions to RT that are not determined by decision formation (e.g., sensory or motor processing). Therefore, RT for a single simulation of the DDM is given by $t_s + ndt$, where $t_s$ is the time at which the bound is reached. Finally, all models included a lapse rate, $\lambda$, which mixes the RT distribution determined by the drift-diffusion process with a uniform distribution in proportion to $\lambda$ (i.e., $\lambda = 0.01$ computes the predicted RT distribution as a weighted average of 99% the DDM distribution and 1% a uniform distribution).

To empirically validate the ability of the DDM to account for our data (the DDM is the continuous-time equivalent of the discrete-time SPRT, which like the generative process in our task is a random-walk process, and one can be used to approximate the other; *Edwards, 1965*; *Smith, 1990*; *Bogacz et al., 2006*), we fit a basic four-parameter DDM ($k_0, B_0, ndt, \lambda$) to each participant's data from the zero-correlation condition. These fits could qualitatively account for the data but were improved by the addition of a collapsing bound (*Figure 6—figure supplement 1*). Therefore, all models in the main analyses included a linear collapsing bound, using parameter $t_B$ to determine the rate of linear collapse. For the *full-$\hat{\rho}$* and *bound-$\hat{\rho}$* models, the bound height at time $t$ is then

$$B_\rho^t = \frac{\sqrt{1 + \hat{\rho}}}{\sqrt{1 + \rho}} \left( B_0 - t_B t \right),$$

such that the correlation-dependent bound adjustment is applied to the instantaneous, and not the initial, bound (in a pilot analysis of data from 22 participants in the 0.6 correlation-magnitude group, we found that the choice of whether to apply this adjustment to the instantaneous or initial bound had a negligible effect on model goodness-of-fit: ΔAIC=−0.9, protected exceedance probability=0.63, in favor of scaling the instantaneous bound over the initial bound). The bounds are symmetric about the starting point, such that $B_\rho^t$ is the distance between the starting point and either bound. Choice commitment occurs when one of the bounds is reached, which happens when $|x(t)| \geq B_\rho^t$, where $x(t)$ is the value of the decision variable at time $t$.

The DDMs were fit to participant's full empirical RT distributions, using PyDDM (*Shinn et al., 2020*). Maximum-likelihood optimization was performed using differential evolution (*Storn and Price, 1997*), a global-optimization algorithm suitable for estimating the parameters of high-dimensional DDMs (*Shinn et al., 2020*). We used the model fits to generate predicted performance for each participant for their actual fit correlation parameters, as well as for the true correlation values and correlations of zero, holding all other parameters fixed at their fit values. We also used PyDDM to generate predictions for the expected performance of an observer that uses the normative form of the bound-height adjustment defined above with $\hat{\rho}$ chosen to explore different levels of correlation underestimation, where $|\rho|$=0.6, and other model parameters were chosen to approximate the average parameters from participants in the 0.6 correlation-magnitude group.

## Data analysis

We conducted statistical analyses in MATLAB (MathWorks) and R (*R Development Core Team, 2020*). We excluded from analysis trials with RTs <0.3 s or >15 s, which are indicative of off-task behavior. This procedure removed 0.8% of the data across participants.

To analyze choice behavior, we fit logistic models to each participant's choices using maximum-likelihood estimation. The basic logistic function was

$$P(R) = \lambda + \frac{1 - 2\lambda}{1 + e^{-\left(\beta_0 + \beta_e \, x \, E[logLR]\right)}},$$

where $P(R)$ is the probability that the subject chose the right source, $\beta_e$ determines the slope of the psychometric function as a function of objective evidence strength (expected logLR), $\beta_0$ is a fixed offset, and $\lambda$ is a lapse rate that sets the lower and upper asymptotes of the logistic curve. We fit two models per participant to assess whether choices were dependent on the correlations: (1) a *joint* model, in which the three free parameters were shared across the three correlation conditions; and (2) a *separate* model, in which a logistic function was fit separately to each correlation condition (nine free parameters).

To assess whether RTs were affected by the correlations, we fit linear mixed-effects models to median RTs per condition, separately for correct and error trials. The predictors included objective evidence strength (low, high), correlation condition ($\rho_-$, 0.0, $\rho_+$), and correlation magnitude (0.2, 0.4, 0.6, 0.8), as well as the interaction between evidence strength and correlation magnitude and the interaction between correlation condition and correlation magnitude. Evidence strength and correlation condition were effect coded, and correlation magnitude was *z*-scored and entered as a continuous covariate. The models were fit using lme4 (*Bates et al., 2015b*). When possible, we fit the maximal model (i.e., random intercepts for subjects and random slopes for all within-subjects variables). In cases where the maximal model failed to converge or yielded singular fits, we iteratively reduced the random-effects structure until convergence (*Bates et al., 2015a*). Significance was assessed via ANOVA using Kenward–Roger *F*-tests with Satterthwaite degrees of freedom, using the car package (*Fox and Weisberg, 2019*).

To assess the relationship between the objective correlations and the subjective fit correlations, we fit linear mixed-effects models to the Fisher-*z*-transformed correlations. To quantify the average deviation of the subjective correlation from the objective correlation, we reversed the signs of the deviations for the negative-correlation conditions so underestimates and overestimates for negative and positive correlations would have the same sign.

## Model comparison

We assessed goodness-of-fit for the logistic and DDMs using Akaike information criteria (AIC). We also used AIC values in a Bayesian random-effects analysis, which attempts to identify the model among competing alternatives that is most frequent in the population. This analysis produced a protected exceedance probability (PEP) for each model, which is the probability that the model is the most frequent in the population, above and beyond chance (*Rigoux et al., 2014*). We computed PEPs using the VBA toolbox (*Daunizeau et al., 2014*).

## Acknowledgements

NT was supported by a T32 training grant from the National Institutes of Health (MH014654). JIG was supported by a CRCNS grant from the National Science Foundation (220727). JK was funded by the Penn Undergraduate Research Mentorship program (PURM). The funders had no role in study design, data collection and analysis, decision to publish, or preparation of the manuscript. We thank Long Ding for helpful comments on the manuscript.

## Additional information

### Competing interests

Joshua I Gold: Senior editor, *eLife*. The other authors declare that no competing interests exist.

### Funding

| Funder | Grant reference number | Author |
|---|---|---|
| National Science Foundation | 220727 | Joshua I Gold |
| National Institutes of Health | 5T32MH014654 | Nathan Tardiff |
| Penn Undergraduate Research Mentorship program (PURM) | | Jiwon Kang |

The funders had no role in study design, data collection and interpretation, or the decision to submit the work for publication.

### Author contributions

Nathan Tardiff, Conceptualization, Data curation, Software, Formal analysis, Supervision, Validation, Investigation, Visualization, Methodology, Writing – original draft, Project administration, Writing – review and editing; Jiwon Kang, Conceptualization, Data curation, Software, Investigation, Methodology, Writing – original draft; Joshua I Gold, Conceptualization, Resources, Software, Formal analysis, Supervision, Funding acquisition, Visualization, Methodology, Writing – original draft, Project administration, Writing – review and editing

### Author ORCIDs

Nathan Tardiff (iD) https://orcid.org/0000-0002-0233-8529
Jiwon Kang (iD) https://orcid.org/0009-0006-7857-3134
Joshua I Gold (iD) https://orcid.org/0000-0002-6018-0483

### Ethics

Participants provided informed consent online via button press. Human protocols were approved and determined to be Exempt by the University of Pennsylvania Internal Review Board (IRB protocol 844474).

Reviewer #1 (Public review): https://doi.org/10.7554/eLife.100258.3.sa1
Reviewer #2 (Public review): https://doi.org/10.7554/eLife.100258.3.sa2
Author response https://doi.org/10.7554/eLife.100258.3.sa3

## Additional files

### Supplementary files
Supplementary file 1. Tables containing average best-fitting parameters for all drift-diffusion models.

MDAR checklist

### Data availability
The datasets generated and analyzed for this article are available at https://osf.io/qygkc/. The analysis code for this article is available at https://github.com/TheGoldLab/Analysis_Tardiff_Kang_Correlated (copy archived at *TheGoldLab, 2024*).

The following dataset was generated:

| Author(s) | Year | Dataset title | Dataset URL | Database and Identifier |
|---|---|---|---|---|
| Tardiff N, Kang J, Gold JI | 2024 | Evidence weighting in uncertain and correlated environments | https://osf.io/qygkc/ | Open Science Framework, qygkc |

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
