## [Editor Report · eLife Assessment]

This **important** work combines theory and experiment to demonstrate **convincingly** how humans make decisions about sequences of pairs of correlated observations. The proposed model for evidence integration in correlated environments will be of use for the study of decision-making.

---

## [Referee Report · Reviewer #1 (Public review)]

Summary:

The behavioral strategies underlying decisions based on perceptual evidence are often studied in the lab with stimuli whose elements provide independent pieces of decision-related evidence that can thus be equally weighted to form a decision. In more natural scenarios, in contrast, the information provided by these pieces is often correlated, which impacts how they should be weighted. Tardiff, Kang & Gold set out to study decisions based on correlated evidence and compare observed behavior of human decision makers to normative decision strategies. To do so, they presented participants with visual sequences of pairs of localized cues whose location was either uncorrelated, or positively or negatively correlated, and whose mean location across a sequence determined the correct choice. Importantly, they adjusted this mean location such that, when correctly weighted, each pair of cues was equally informative, irrespective of how correlated it was. Thus, if participants follow the normative decision strategy, their choices and reaction times should not be impacted by these correlations. While Tardiff and colleagues found no impact of correlations on choices, they did find them to impact reaction times, suggesting that participants deviated from the normative decision strategy. To assess the degree of this deviation, Tardiff et al. adjusted drift diffusion models (DDMs) for decision-making to process correlated decision evidence. These fits, and a comparison of different model variants revealed that participants considered correlations when weighing evidence, but did so with a slight underestimation of magnitude of this correlation. This finding made Tardiff et al. conclude that participants followed a close-to normative decision strategy that adequately took into account correlated evidence.

Strength:

The authors adjust a previously used experimental design to include correlated evidence in a simple, yet powerful way. The way it does so is easy to understand and intuitive, such that participants don't need extensive training to perform the task. Limited training makes it more likely that the observed behavior is natural and reflective of every-day decision-making. Furthermore, the design allowed the authors to make the amount of decision-related evidence equal across different correlation magnitudes, which makes it easy to assess whether participants correctly take account of these correlations when weighing evidence: if they do, their behavior should not be impacted by the correlation magnitude.

The relative simplicity with which correlated evidence is introduced also allowed the authors to fall back to the well-established DDM for perceptual decisions, that has few parameters, is known to implement the normative decision strategy in certain circumstances, and enjoys a great deal of empirical support. The authors show how correlations ought to impact these parameters, and which changes in parameters one would expect to see if participants mis-estimate these correlations or ignore them altogether (i.e., estimate correlations to be zero). This allowed them to assess the degree to which participants took into account correlations on the full continuum from perfect evidence weighting to complete ignorance. More specifically, the authors showed that a consistent mis-estimation of the correlation magnitude would not impact the fraction of correct choices (as they observe), but only the reaction times. With this, they could show that participants in fact performed rational evidence weighting if one assumed that they slightly underestimated the correlation magnitude.

Weaknesses:

While the authors convincingly demonstrate that the observed decision-making behavior seems to stem from a slight underestimation of the correlation magnitudes, their experimental paradigm did not allow them to determine the origin of this bias. Through additional analyses they rule out various possibilities, like the impact of a Bayesian prior on estimated correlations. Nonetheless, the authors provide no normative explanation of the observed bias.

A further minor weakness is that the authors only focus on a single normative aspect of the observed behavior, namely on whether participants optimally accumulate decision-related evidence across time. Another question is whether participants tune their decision boundaries to maximize reward rates or some other overall performance measures. While the authors discuss that the chosen diffusion models (DDMs) have the potential of also implementing normative decisions in the latter sense, the authors' analysis does not address this question in the context of their task.

---

## [Referee Report · Reviewer #2 (Public review)]

This study by Tardiff, Kang & Gold seeks to (i) develop a normative account of how observers should adapt their decision-making across environments with different levels of correlation between successive pairs of observations, and (ii) assess whether human decisions in such environments are consistent with this normative model. The authors first demonstrate that, in the range of environments under consideration here, an observer with full knowledge of the generative statistics should take both the magnitude and sign of the underlying correlation into account when assigning weight in their decisions to new observations: stronger negative correlations should translate into stronger weighting (due to the greater information furnished by an anticorrelated generative source), while stronger positive correlations should translate into weaker weighting (due to the greater redundancy of information provided by a positively correlated generative source). The authors then report an empirical study in which human participants performed a perceptual decision-making task requiring accumulation of information provided by pairs of perceptual samples, under different levels of pairwise correlation. They describe a nuanced pattern of results with effects of correlation being largely restricted to response times and not choice accuracy, which could be captured through fits of their normative model (in this implementation, an extension of the well-known drift diffusion model) to the participants' behaviour while allowing for mis-estimation of the underlying correlations. An intriguing result is that the observed pattern of behavioural effects is best explained by a model in which observers marginally underestimated the level of correlation between the generative sources, and that this bias affects behaviour through effects on stimulus encoding that then shape how the evidence furnished by each stimulus sample is weighted in decision formation.

As the authors point out in their very well-written paper, appropriate weighting of information gathered in correlated environments has important consequences for real-world decision-making. Yet, while this function has been well studied for 'high-level' (e.g. economic) decisions, how we account for correlations when making simple perceptual decisions on well-controlled behavioural tasks has not been investigated. As such, this study addresses an important and timely question that will be of broad interest to psychologists and neuroscientists. The computational approach to arrive at normative principles for evidence weighting across environments with different levels of correlation is elegant, makes strong connections with prior work in different decision-making contexts, and should serve as a valuable reference point for future studies in this domain. The empirical study is well designed and executed, and the modelling approach applied to these data showcases an impressively deep understanding of relationships between different parameters of the drift diffusion model and its novel application to this setting. Another strength of the study is that it is preregistered.

In my view, any major weaknesses of the study have been well addressed by the authors during review. An outstanding question that arises from the current work and remains unanswered here is around the (normative?) origin of the correlation underestimates, and the present work lays a strong foundation from which to pursue this question in the future.

---

## [Author Response]

The following is the authors’ response to the original reviews

We thank the reviewers for their thoughtful feedback. We have made substantial revisions to the manuscript to address each of their comments, as we detail below. We want to highlight one major change in particular that addresses a concern raised by both reviewers: the role of the drift rate in our models. Motivated by their astute comments, we went back through our models and realized that we had made a particular assumption that deserved more scrutiny. We previously assumed that the process of encoding the observations made correct use of the objective, generative correlation, but then the process of calculating the weight of evidence used a mis-scaled, subjective version of the correlation. These assumptions led us to scale the drift rate in the model by a term that quantified how the standard deviation of the observation distribution was affected by the objective correlation (encoding), but to scale the bound height by the subjective estimate of the correlation (evidence weighing). However, we realized that encoding may also depend on the subjective correlation experienced by the participant. We have now tested several alternative models and found that the best-fitting model assumes that a single, subjective estimate of the correlation governs both encoding and evidence weighing. An important consequence of updating our models in this way is that we can now account for the behavioral data without needing the additional correlation-dependent drift terms (which, as reviewer #2 pointed out, were difficult to explain).

We also note that we changed the title slightly, replacing “weighting” with “weighing” for consistency with our usage throughout the manuscript.

Please see below for more details about this important point and our responses to the reviewers’ specific concerns.

**Public Reviews:**

**Reviewer #1 (Public review):**
Summary:The behavioral strategies underlying decisions based on perceptual evidence are often studied in the lab with stimuli whose elements provide independent pieces of decision-related evidence that can thus be equally weighted to form a decision. In more natural scenarios, in contrast, the information provided by these pieces is often correlated, which impacts how they should be weighted. Tardiff, Kang & Gold set out to study decisions based on correlated evidence and compare the observed behavior of human decision-makers to normative decision strategies. To do so, they presented participants with visual sequences of pairs of localized cues whose location was either uncorrelated, or positively or negatively correlated, and whose mean location across a sequence determined the correct choice. Importantly, they adjusted this mean location such that, when correctly weighted, each pair of cues was equally informative, irrespective of how correlated it was. Thus, if participants follow the normative decision strategy, their choices and reaction times should not be impacted by these correlations. While Tardiff and colleagues found no impact of correlations on choices, they did find them to impact reaction times, suggesting that participants deviated from the normative decision strategy. To assess the degree of this deviation, Tardiff et al. adjusted drift-diffusion models (DDMs) for decision-making to process correlated decision evidence. Fitting these models to the behavior of individual participants revealed that participants considered correlations when weighing evidence, but did so with a slight underestimation of the magnitude of this correlation. This finding made Tardiff et al. conclude that participants followed a close-to-normative decision strategy that adequately took into account correlated evidence.Strengths:The authors adjust a previously used experimental design to include correlated evidence in a simple, yet powerful way. The way it does so is easy to understand and intuitive, such that participants don't need extensive training to perform the task. Limited training makes it more likely that the observed behavior is natural and reflective of everyday decision-making. Furthermore, the design allowed the authors to make the amount of decision-related evidence equal across different correlation magnitudes, which makes it easy to assess whether participants correctly take account of these correlations when weighing evidence: if they do, their behavior should not be impacted by the correlation magnitude.The relative simplicity with which correlated evidence is introduced also allowed the authors to fall back to the well-established DDM for perceptual decisions, which has few parameters, is known to implement the normative decision strategy in certain circumstances, and enjoys a great deal of empirical support. The authors show how correlations ought to impact these parameters, and which changes in parameters one would expect to see if participants misestimate these correlations or ignore them altogether (i.e., estimate correlations to be zero). This allowed them to assess the degree to which participants took into account correlations on the full continuum from perfect evidence weighting to complete ignorance. With this, they could show that participants in fact performed rational evidence weighting if one assumed that they slightly underestimated the correlation magnitude.

Weaknesses:

The experiment varies the correlation magnitude across trials such that participants need to estimate this magnitude within individual trials. This has several consequences:

(1) Given that correlation magnitudes are estimated from limited data, the (subjective) estimates might be biased towards their average. This implies that, while the amount of evidence provided by each 'sample' is objectively independent of the correlation magnitude, it might subjectively depend on the correlation magnitude. As a result, the normative strategy might differ across correlation magnitudes, unlike what is suggested in the paper. In fact, it might be the case that the observed correlation magnitude underestimates corresponds to the normative strategy.

We thank the reviewer for raising this interesting point, which we now address directly with new analyses including model fits (pp. 15–24). These analyses show that the participants were computing correlation-dependent weights of evidence from observation distributions that reflected suboptimal misestimates of correlation magnitudes. This strategy is normative in the sense that it is the best that they can do, given the encoding suboptimality. However, as we note in the manuscript, we do not know the source of the encoding suboptimality (pp. 23–24). We thus do not know if there might be a strategy they could have used to make the encoding more optimal.

(2) The authors link the normative decision strategy to putting a bound on the log-likelihood ratio (logLR), as implemented by the two decision boundaries in DDMs. However, as the authors also highlight in their discussion, the 'particle location' in DDMs ceases to correspond to the logLR as soon as the strength of evidence varies across trials and isn't known by the decision maker before the start of each trial. In fact, in the used experiment, the strength of evidence is modulated in two ways:(i) by the (uncorrected) distance of the cue location mean from the decision boundary (what the authors call the evidence strength) and(ii) by the correlation magnitude. Both vary pseudo-randomly across trials, and are unknown to the decision-maker at the start of each trial. As previous work has shown (e.g. Kiani & Shadlen (2009), Drugowitsch et al. (2012)), the normative strategy then requires averaging over different evidence strength magnitudes while forming one's belief. This averaging causes the 'particle location' to deviate from the logLR. This deviation makes it unclear if the DDM used in the paper indeed implements the normative strategy, or is even a good approximation to it.

We appreciate this subtle, but important, point. We now clarify that the DDM we use includes degrees of freedom that are consistent with normative decision processes that rely on the imperfect knowledge that participants have about the generative process on each trial, specifically: (1) a single drift-rate parameter that is fit to data across different values of the mean of the generative distribution, which is based on the standard assumption for these kinds of task conditions in which stimulus strength is varied randomly from trial-to-trial and thus prevents the use of exact logLR (which would require stimulus strength-specific scale factors; Gold and Shadlen, 2001); (2) the use of a collapsing bound, which in certain cases (including our task) is thought to support a stimulus strength-dependent calibration of the decision variable to optimize decisions (Drugowitsch et al, 2012); and (3) free parameters (one per correlation) to account for subjective estimates of the correlation, which affected the encoding of the observations that are otherwise weighed in a normative manner in the best-fitting model.

Also, to clarify our terminology, we define the objective evidence strength as the expected logLR in a given condition, which for our task is dependent on both the distance of the mean from the decision boundary and the correlation (p. 7).

Given that participants observe 5 evidence samples per second and on average require multiple seconds to form their decisions, it might be that they are able to form a fairly precise estimate of the correlation magnitude within individual trials. However, whether this is indeed the case is not clear from the paper.

These points are now addressed directly in Results (pp. 23–24) and Figure 7 supplemental figures 1–3. Specifically, we show that, as the reviewer correctly surmised above, empirical correlations computed on each trial tended to be biased towards zero (Fig 7–figure supplement 1). However, two other analyses were not consistent with the idea that participants’ decisions were based on trial-by-trial estimates of the empirical correlations: (1) those with the shortest RTs did not have the most-biased estimates (Fig 7–figure supplement 2), and (2) there was no systematic relationship between objective and subjective fit correlations across participants (Fig 7–figure supplement 3).

Furthermore, the authors capture any underestimation of the correlation magnitude by an adjustment to the DDM bound parameter. They justify this adjustment by asking how this bound parameter needs to be set to achieve correlation-independent psychometric curves (as observed in their experiments) even if participants use a 'wrong' correlation magnitude to process the provided evidence. Curiously, however, the drift rate, which is the second critical DDM parameter, is not adjusted in the same way. If participants use the 'wrong' correlation magnitude, then wouldn't this lead to a mis-weighting of the evidence that would also impact the drift rate? The current model does not account for this, such that the provided estimates of the mis-estimated correlation magnitudes might be biased.

We appreciate this valuable comment, and we agree that we previously neglected the potential impact of correlation misestimates on evidence strength. As we now clarify, the correlation enters these models in two ways: (1) via its effect on how the observations are encoded, which involves scaling both the drift and the bound; and (2) via its effect on evidence weighing, which involves scaling only the bound (pp. 15–18). We previously assumed that only the second form of scaling might involve a subjective (mis-)estimate of the correlation. We now examine several models that also include the possibility of either or both forms using subjective correlation estimates. We show that a model that assumes that the same subjective estimate drives both encoding and weighing (the “full-rho-hat” model) best accounts for the data. This model provides better fits (after accounting for differences in numbers of parameters) than models with: (1) no correlation-dependent adjustments (“base” model), (2) separate drift parameters for each correlation condition (“drift” model), (3) optimal (correlation-dependent) encoding but suboptimal weighing (“bound-rho-hat” model, which was our previous formulation), (4) suboptimal encoding and weighing (“scaled-rho-hat” model), and (5) optimal encoding but suboptimal weighing and separate correlation-dependent adjustments to the drift rate (“boundrho-hat plus drift” model). We have substantially revised Figures 5–7 and the associated text to address these points.

Lastly, the paper makes it hard to assess how much better the participants' choices would be if they used the correct correlation magnitudes rather than underestimates thereof. This is important to know, as it only makes sense to strictly follow the normative strategy if it comes with a significant performance gain.

We now include new analyses in Fig. 7 that demonstrate how much participants' choices and RT deviate from: (1) an ideal observer using the objective correlations, and (2) an observer who failed to adjust for the fit subjective correlation when weighing the evidence (i.e., using the subjective correlation for encoding but a correlation of zero for weighing). We now indicate that participants’ performance was quite close to that predicted by the ideal observer (using the true, objective correlation) for many conditions. Thus, we agree that they might not have had the impetus to optimize the decision process further, assuming it were possible under these task conditions.

**Reviewer #2 (Public review):**
Summary:This study by Tardiff, Kang & Gold seeks to: (i) develop a normative account of how observers should adapt their decision-making across environments with different levels of correlation between successive pairs of observations, and (ii) assess whether human decisions in such environments are consistent with this normative model.The authors first demonstrate that, in the range of environments under consideration here, an observer with full knowledge of the generative statistics should take both the magnitude and sign of the underlying correlation into account when assigning weight in their decisions to new observations: stronger negative correlations should translate into stronger weighting (due to the greater information furnished by an anticorrelated generative source), while stronger positive correlations should translate into weaker weighting (due to the greater redundancy of information provided by a positively correlated generative source). The authors then report an empirical study in which human participants performed a perceptual decision-making task requiring accumulation of information provided by pairs of perceptual samples, under different levels of pairwise correlation. They describe a nuanced pattern of results with effects of correlation being largely restricted to response times and not choice accuracy, which could partly be captured through fits of their normative model (in this implementation, an extension of the well-known drift-diffusion model) to the participants' behaviour while allowing for misestimation of the underlying correlations.Strengths:As the authors point out in their very well-written paper, appropriate weighting of information gathered in correlated environments has important consequences for real-world decisionmaking. Yet, while this function has been well studied for 'high-level' (e.g. economic) decisions, how we account for correlations when making simple perceptual decisions on well-controlled behavioural tasks has not been investigated. As such, this study addresses an important and timely question that will be of broad interest to psychologists and neuroscientists. The computational approach to arrive at normative principles for evidence weighting across environments with different levels of correlation is very elegant, makes strong connections with prior work in different decision-making contexts, and should serve as a valuable reference point for future studies in this domain. The empirical study is well designed and executed, and the modelling approach applied to these data showcases a deep understanding of relationships between different parameters of the drift-diffusion model and its application to this setting. Another strength of the study is that it is preregistered.Weaknesses:In my view, the major weaknesses of the study center on the narrow focus and subsequent interpretation of the modelling applied to the empirical data. I elaborate on each below:Modelling interpretation: the authors' preference for fitting and interpreting the observed behavioural effects primarily in terms of raising or lowering the decision bound is not well motivated and will potentially be confusing for readers, for several reasons. First, the entire study is conceived, in the Introduction and first part of the Results at least, as an investigation of appropriate adjustments of evidence weighting in the face of varying correlations. The authors do describe how changes in the scaling of the evidence in the drift-diffusion model are mathematically equivalent to changes in the decision bound - but this comes amidst a lengthy treatment of the interaction between different parameters of the model and aspects of the current task which I must admit to finding challenging to follow, and the motivation behind shifting the focus to bound adjustments remained quite opaque.

We appreciate this valuable feedback. We have revised the text in several places to make these important points more clearly. For example, in the Introduction we now clarify that “The weight of evidence is computed as a scaled version of each observation (the scaling can be applied to the observations or to the bound, which are mathematically equivalent; Green and Swets, 1966) to form the logLR” (p. 3). We also provide more details and intuition in the Results section for how and why we implemented the DDM the way we did. In particular, we now emphasize that the correlation enters these models in two ways: (1) via its effect on encoding the observations, which scales both the drift and the bound; and (2) via its effect on evidence weighing, which scales only the bound (pp. 15–18).

Second, and more seriously, bound adjustments of the form modelled here do not seem to be a viable candidate for producing behavioural effects of varying correlations on this task. As the authors state toward the end of the Introduction, the decision bound is typically conceived of as being "predefined" - that is, set before a trial begins, at a level that should strike an appropriate balance between producing fast and accurate decisions. There is an abundance of evidence now that bounds can change over the course of a trial - but typically these changes are considered to be consistently applied in response to learned, predictable constraints imposed by a particular task (e.g. response deadlines, varying evidence strengths). In the present case, however, the critical consideration is that the correlation conditions were randomly interleaved across trials and were not signaled to participants in advance of each trial - and as such, what correlation the participant would encounter on an upcoming trial could not be predicted. It is unclear, then, how participants are meant to have implemented the bound adjustments prescribed by the model fits. At best, participants needed to form estimates of the correlation strength/direction (only possible by observing several pairs of samples in sequence) as each trial unfolded, and they might have dynamically adjusted their bounds (e.g. collapsing at a different rate across correlation conditions) in the process. But this is very different from the modelling approach that was taken. In general, then, I view the emphasis on bound adjustment as the candidate mechanism for producing the observed behavioural effects to be unjustified (see also next point).

We again appreciate this valuable feedback and have made a number of revisions to try to clarify these points. In addition to addressing the equivalence of scaling the evidence and the bound in the Introduction, we have added the following section to Results (Results, p.18):

“Note that scaling the bound in these formulations follows conventions of the DDM, as detailed above, to facilitate interpretation of the parameters. These formulations also raise an apparent contradiction: the “predefined” bound is scaled by subjective estimates of the correlation, but the correlation was randomized from trial to trial and thus could not be known in advance. However, scaling the bound in these ways is mathematically equivalent to using a fixed bound on each trial and scaling the observations to approximate logLR (see Methods). This equivalence implies that in the brain, effectively scaling a “predefined” bound could occur when assigning a weight of evidence to the observations as they are presented.”

We also note in Methods (pp. 40–41):

“In the DDM, this scaling of the evidence is equivalent to assuming that the decision variable accumulates momentary evidence of the form (x1 + x2) and then dividing the bound height by the appropriate scale factor. An alternative approach would be to scale both the signal and noise components of the DDM by the scale factor. However, scaling the bound is both simpler and maintains the conventional interpretation of the DDM parameters in which the bound reflects the decision-related components of the evidence accumulation process, and the drift rate represents sensory-related components.”

We believe we provide strong evidence that participants adjust their evidence weighing to account for the correlations (see response below), but we remain agnostic as to how exactly this weighing is implemented in the brain.

Modelling focus: Related to the previous point, it is stated that participants' choice and RT patterns across correlation conditions were qualitatively consistent with bound adjustments (p.20), but evidence for this claim is limited. Bound adjustments imply effects on both accuracy and RTs, but the data here show either only effects on RTs, or RT effects mixed with accuracy trends that are in the opposite direction to what would be expected from bound adjustment (i.e. slower RT with a trend toward diminished accuracy in the strong negative correlation condition; Figure 3b). Allowing both drift rate and bound to vary with correlation conditions allowed the model to provide a better account of the data in the strong correlation conditions - but from what I can tell this is not consistent with the authors' preregistered hypotheses, and they rely on a posthoc explanation that is necessarily speculative and cannot presently be tested (that the diminished drift rates for higher negative correlations are due to imperfect mapping between subjective evidence strength and the experimenter-controlled adjustment to objective evidence strengths to account for effects of correlations). In my opinion, there are other candidate explanations for the observed effects that could be tested but lie outside of the relatively narrow focus of the current modelling efforts. Both explanations arise from aspects of the task, which are not mutually exclusive. The first is that an interesting aspect of this task, which contrasts with most common 'univariate' perceptual decision-making tasks, is that participants need to integrate two pieces of information at a time, which may or may not require an additional computational step (e.g. averaging of two spatial locations before adding a single quantum of evidence to the building decision variable). There is abundant evidence that such intermediate computations on the evidence can give rise to certain forms of bias in the way that evidence is accumulated (e.g. 'selective integration' as outlined in Usher et al., 2019, Current Directions in Psychological Science; Luyckx et al., 2020, Cerebral Cortex) which may affect RTs and/or accuracy on the current task. The second candidate explanation is that participants in the current study were only given 200 ms to process and accumulate each pair of evidence samples, which may create a processing bottleneck causing certain pairs or individual samples to be missed (and which, assuming fixed decision bounds, would presumably selectively affect RT and not accuracy). If I were to speculate, I would say that both factors could be exacerbated in the negative correlation conditions, where pairs of samples will on average be more 'conflicting' (i.e. further apart) and, speculatively, more challenging to process in the limited time available here to participants. Such possibilities could be tested through, for example, an interrogation paradigm version of the current task which would allow the impact of individual pairs of evidence samples to be more straightforwardly assessed; and by assessing the impact of varying inter-sample intervals on the behavioural effects reported presently.

We thank the reviewer for this thoughtful and valuable feedback. We have thoroughly updated the modeling section to include new analysis and clearer descriptions and interpretations of our findings (including Figs. 5–7 and additional references to the Usher, Luyckx, and other studies that identified decision suboptimalities). The comment about “an additional computational step” in converting the observations to evidence was particularly useful, in that it made us realize that we were making what we now consider to be a faulty assumption in our version of the DDM. Specifically, we assumed that subjective misestimates of the correlation affected how observations were converted to evidence (logLR) to form the decision (implemented as a scaling of the bound height), but we neglected to consider how suboptimalities in encoding the observations could also lead to misestimates of the correlation. We have retained the previous best-fitting models in the text, for comparison (the “bound-rho-hat” and “bound-rho-hat + drift” models). In addition, we now include a “full-rho-hat” model that assumes that misestimates of rho affect both the encoding of the observations, which affects the drift rate and bound height, and the weighing of the evidence, which affects only the bound height. This was the best-fitting model for most participants (after accounting for different numbers of parameters associated with the different models we tested). Note that the full-rho-hat model predicts the lack of correlation-dependent choice effects and the substantial correlation-dependent RT effects that we observed, without requiring any additional adjustments to the drift rate (as we resorted to previously).

In summary, we believe that we now have a much more parsimonious account of our data, in terms of a model in which subjective estimates of the correlation are alone able to account for our patterns of choice and RT data. We fully agree that more work is needed to better understand the source of these misestimates but also think those questions are outside the scope of the present study.

**Recommendations for the authors:**

**Reviewer #1 (Recommendations for the authors):**
A few minor comments:(1) Evidence can be correlated in multiple ways. It could be correlated within individual pieces of evidence in a sequence, or across elements in that sequence (e.g., across time). This distinction is important, as it determines how evidence ought to be accumulated across time. In particular, if evidence is correlated across time, simply summing it up might be the wrong thing to do. Thus, it would be beneficial to make this distinction in the Introduction, and to mention that this paper is only concerned with the first type of correlation.

We now clarify this point in the Introduction (p. 5–6).

(2) It is unclear without reading the Methods how the blue dashed line in Figure 4c is generated. To my understanding, it is a prediction of the naive DDM model. Is this correct?

We now specify the models used to make the predictions shown in Fig. 4c (which now includes an additional model that uses unscaled observations as evidence).

(3) In Methods, given the importance of the distribution of x1 + x2, it would be useful to write it out explicitly, e.g., x1 + x2 ~ N(2 mu_g, ..), specifying its mean and its variance.

Excellent suggestion, added to p. 38.

(4) From Methods and the caption of Figure 6 - Supplement 1 it becomes clear that the fitted DDM features a bound that collapses over time. I think that this should also be mentioned in the main text, as it is a not-too-unimportant feature of the model.

Excellent suggestion, added to p. 15, with reference to Fig. 6-supplement 1 on p. 20.

(5) The functional form of the bound is 2 (B - tb t). To my understanding, the effective B changes as a function of the correlation magnitude. Does tb as well? If not, wouldn't it be better if it does, to ensure that 2 (B - tb t) = 0 independent of the correlation magnitude?

In our initial modeling, we also considered whether the correlation-dependent adjustment, which is a function of both correlation sign and magnitude, should be applied to the initial bound or to the instantaneous bound (i.e., after collapse, affecting tb as well). In a pilot analysis of data from 22 participants in the 0.6 correlation-magnitude group, we found that this choice had a negligible effect on the goodness-of-fit (deltaAIC = -0.9, protected exceedance probability = 0.63, in favor of the instantaneous bound scaling). We therefore used the instantaneous bound version in the analyses reported in the manuscript but doubt this choice was critical based on these results. We have clarified our implementation of the bound in Methods (p. 43–44).

**Reviewer #2 (Recommendations for the authors):**
In addition to the points raised above, I have some minor suggestions/open questions that arose from my reading of the manuscript:(1) Are the predictions outlined in the paper specific to cases where the two sources are symmetric around zero? If distributions are allowed to be asymmetric then one can imagine cases (i.e. when distribution means are sufficiently offset from one another) where positive correlations can increase evidence strength and negative correlations decrease evidence strength. There's absolutely still value and much elegance in what the authors are showing with this work, but if my intuition is correct, it should ideally be acknowledged that the predictions are restricted to a specific set of generative circumstances.

We agree that there are a lot of ways to manipulate correlations and their effect on the weight of evidence. At the end of the Discussion, we emphasize that our results apply to this particular form of correlation (p. 32).

(2) Isn't Figure 4C misleading in the sense that it collapses across the asymmetry in the effect of negative vs positive correlations on RT, which is clearly there in the data and which simply adjusting the correlation-dependent scale factor will not reproduce?

We agree that this analysis does not address any asymmetries in suboptimal estimates of positive versus negative correlations. We believe that those effects are much better addressed using the model fitting, which we present later in the Results section. We have now simplified the analyses in Fig. 4c, reporting the difference in RT between positive and negative correlation conditions instead of a linear regression.

(3) I found the transition on p.17 of the Results section from the scaling of drift rate by correlation to scaling of bound height to be quite abrupt and unclear. I suspect that many readers coming from a typical DDM modelling background will be operating under the assumption that drift rate and bound height are independent, and I think more could be done here to explain why scaling one parameter by correlation in the present case is in fact directly equivalent to scaling the other.

Thank you for the very useful feedback, we have substantially revised this text to make these points more clearly.

(4) P.3, typo: Alan *Turing*

That’s embarrassing. Fixed.

(5) P.27, typo: "participants adopt a *fixed* bound"

Fixed.